# Application of the Complete Data Fusion to the ozone profiles measured by geostationary and low earth orbits satellites: a feasibility study

Nicola Zoppetti[1], Simone Ceccherini[1], Bruno Carli[1], Samuele Del Bianco[1], Marco Gai[1], Cecilia Tirelli[1], Flavio Barbara[1], Rossana Dragani[2], Antti Arola[3], Jukka Kujanpää[4], Jacob C.A. van Peet[5,6], Ronald van der A[5] and Ugo Cortesi[1]

[1] Istituto di Fisica Applicata "Nello Carrara" del Consiglio Nazionale delle Ricerche, Via Madonna del Piano 10, 50019 Sesto Fiorentino, Italy
[2] European Centre for Medium-Range Weather Forecasts, Shinfield Park, Reading, RG2 9AX, UK
[3] Finnish Meteorological Institute, Atmospheric Research Centre of Eastern Finland, P.O.Box 1627, 70211 Kuopio, Finland
[4] Finnish Meteorological Institute, Space and Earth Observation Centre, P.O. Box 503, FI-00101 Helsinki, Finland
[5] Royal Netherlands Meteorological Institute, Utrechtseweg 297, 3731 GA De Bilt, The Netherlands
[6] Vrije Universiteit Amsterdam, Department of Earth Sciences, Amsterdam, The Netherlands

*Correspondence to*: Nicola Zoppetti (N.Zoppetti@ifac.cnr.it)

**Abstract.** The new platforms for Earth observation from space are characterized by measurements made with great spatial and temporal resolution. While this abundance of information makes it possible to detect and study localized phenomena, on the other hand, it may be difficult to manage this large amount of data in the study of global and large-scale phenomena.

A particularly significant example is the use by assimilation systems of Level 2 products that represent gas profiles in the atmosphere. The models on which assimilation systems are based are discretized on spatial grids with horizontal dimensions of the order of tens of kilometres in which tens or hundreds of measurements may fall in the future.

A simple procedure to overcome this problem is to extract a subset of the original measurements but this involves a loss of information; another option is the use of simple averages of the profiles, but also this approach has some limitations that we will discuss in the paper. A more advanced solution is to resort to the so-called fusion algorithms, capable of compressing the size of the dataset while limiting the information loss. A novel data fusion method, the Complete Data Fusion, was recently developed to merge a posteriori a set of retrieved products in a single product. In the present paper, we apply the Complete Data Fusion method to ozone profile measurements simulated in the thermal infrared and ultraviolet bands, in a realistic scenario. Then the fused products are compared with the input profiles; comparisons show that the output products of data fusion have in general smaller total errors and higher information contents. The comparisons of the fused with the fusing products are presented both at single fusion grid-box scale and with a statistical analysis of the results obtained on large sets of fusion grid-box of the same size. We also evaluate the grid box size impact, showing that the Complete Data Fusion method can be used with different grid-box sizes even if this possibility is connected to the natural variability of the considered atmospheric molecule.

## 1. Introduction

In the context of the Copernicus programme (https://www.copernicus.eu) coordinated by the European Commission, the European Space Agency is responsible for the Space Component consisting of a novel set of Earth Observation (EO) satellite missions for environmental monitoring applications: the Sentinels (https://sentinel.esa.int/web/sentinel/missions). Each mission focuses on a specific aspect of EO. In particular, the geostationary (GEO) mission Sentinel-4 and the two Low Earth Orbit (LEO) missions (Sentinel-5p and Sentinel 5), referred to as the atmospheric Sentinels, are dedicated to monitoring air quality, stratospheric ozone, ultraviolet surface radiation and climate.

The atmospheric Sentinels will provide an enormous amount of data with unprecedented accuracy and spatio-temporal resolution. In this scenario, a central challenge is to enable a generic data user (for example, an assimilation system) to exploit such a large amount of data.

A variety of approaches can serve the purpose to convey in a single product the information associated with remote sensing observations of the vertical distribution of a given atmospheric target from multiple independent sources. Strategies for the combined use of multiple atmospheric profile datasets include a posteriori data fusion techniques, synergistic inversion processes (Aires et al., 2012 and references therein; Natraj et al., 2011; Cuesta et al., 2013; Cortesi et al., 2016; Sato et al., 2018) and, in broader terms, might include assimilation systems (Lahoz and Schneider, 2014).

The three approaches differ in the accepted inputs and in the involved models. In the synergistic inversion, the inputs consist of the radiance observations (Level 1 products) of all the measurements and the output profiles are obtained by a simultaneous retrieval of these observations. A posteriori fusion techniques consist of sophisticated averaging processes in which the inputs are profiles (Level 2 products) retrieved from the single measurements. The assimilation techniques, in their more general implementations, can accept as inputs both radiances and profiles and use the information of the measurements as inputs of an atmospheric model. Each of these strategies implies different advantages and drawbacks, ultimately assessing the cost-to-benefit ratio that drives the selection of the option of choice for the specific case under investigation.

In particular Data fusion algorithms, such as the Complete Data Fusion (CDF) (Ceccherini et al., 2015), can be well suited to reduce the data volume that users need to access and handle while retaining the information content of the whole level 2 (L2) products.

The CDF inputs are any number of L2 profiles retrieved with the optimal estimation technique and characterized by their a priori information, covariance matrix (CM) and averaging kernel (AK) matrix. The output of the CDF is a single product (also characterized by an a priori, a CM and AK matrices) in which the vertical sensitivity increases and the error reduces with respect to the inputs (Ceccherini et al., 2015).

This work is based on the simulated data produced in the context of the Advanced Ultraviolet Radiation and Ozone Retrieval for Applications project (AURORA, Cortesi et al.,2018), funded by the European Commission in the framework of the Horizon 2020 programme. The project regards the sequential application of fusion and assimilation algorithms to ozone profiles simulated according to specifications similar to those of the atmospheric Sentinels.

The use of synthetic data allows evaluating the performances of the algorithms in terms of differences between the products and a reference truth, represented by the atmospheric scenario used in the L2 simulation procedure. On the other hand, the absence of systematic errors in the simulated measurements limits the study to ideal measurement conditions. However, the CDF algorithm intrinsically provides a mechanism to include different kinds of errors into the analysis. For instance, Ceccherini et al. (2018) discussed how to treat interpolation and coincidence errors, while Ceccherini et al. (2019) explicitly introduce the treatment of systematic errors.

This work is divided into two parts. In the first part, we describe the datasets and methodologies (the L2 simulation procedure and the CDF) and discuss the differences between CDF and mere averaging. In the second part, the quality of the fused products obtained from L2 profiles that are not perfectly co-located in space and in time is analysed. To account for the geo-temporal differences in the L2 profiles, a coincidence error is added to the fused product error budget. The fused and standard L2 products are compared and assessed in terms of their information content, highlighting the better data quality provided by the fusion. Finally, we also show that the CDF can be applied with different coincidence grid-box sizes, allowing for different compression factors of the Level 2 input data volume.

Some of the characteristics of the products used in this work differ from what they will be in reality: in particular the spatial sampling (spacing between pixels and shape) and in some cases the signal-to-noise ratio (GEO-TIR instrument) of the instruments used in the present paper (both GEO and LEO) are different from those of that will be onboard the Sentinel 4/MTG-

S and EPS-SG. Nevertheless, this work focuses on a comparison of the fused and L2 products and, in particular, on the ability of the CDF inducing quality improvements that are, in some sense, independent from precise instrumental characteristics. The application of CDF to L2 products simulated with the characteristics even only similar to the one expected from the atmospheric Sentinel 4 and 5 allows establishing the possible benefits in case of real Sentinel data.

## 2. Material and methods

### 2.1. Atmospheric scenario and ozone climatology

In this work, we used two basic external sources to generate the database of the standard L2 ozone products: the ozone climatology and the atmospheric scenario.

We used the ozone climatology as a priori information in both the simulation of L2 products and the CDF. The atmospheric scenario represents the true state of the atmosphere, and we used it in both the simulation of L2 products and the quality assessment of the fused ones.

In particular, the ozone climatology was derived from McPeters and Labow (McPeters and Labow, 2012) and directly provided the a priori profile $x_a$ used either in the simulation expressions (Eq. (1)), as well as in the fusion (see Eq. (6)). We calculate the diagonal terms of the a priori CM $\mathbf{S}_a$ as the square of the standard deviation of McPeters and Labow climatology, putting an inferior limit to this diagonal value equal to the square of 20% of the a priori profile. The off-diagonal elements are calculated using a correlation length of 6 km. The correlation length is used to reduce oscillations in the simulated profiles, and 6 km is the typical value used for nadir ozone profile retrieval (Liu et al., 2010, Miles et al., 2015). The a priori CM is used in the in the expression of the L2 AK matrix (Eq. (2)) and in Eq. (4) and (5) of the next paragraph. The a priori CM $\mathbf{S}_a$ plays an important role also in the CDF equations (see Eq. (6)).

The atmospheric scenario is taken from the Modern Era-Retrospective analysis for Research and Applications version 2 (MERRA2) reanalysis (Gelaro et al., 2017). The MERRA2 data are provided by the Global Modelling and Assimilation Office (GMAO) at NASA Goddard Space Flight Center. This reanalysis covers the recent time of remotely sensed data, from 1979 through the present. The atmospheric scenario is the source of true profile $x_t$ used in Eq. (1) to synthesize the simulated L2 products and represents the main reference for the comparison of the quality of L2 and fused products.

### 2.2. L2 Product Simulation Algorithm

The simulation algorithm has been originally formalized in the context of the AURORA project, aiming at an efficient computational process. The L2 retrieved state is simulated on a fixed vertical grid with a 3 km step, by the linear approximation given in Eq. (1):

$$\hat{x} = \mathbf{A}x_t + (\mathbf{I} - \mathbf{A})x_a + \boldsymbol{\delta} \tag{1}$$

In Eq. (1), $x_t$ is the true state of the atmosphere represented by the atmospheric scenarios, $x_a$ is the a priori estimate of the state vector provided by the ozone climatology, $\boldsymbol{\delta}$ is the uncertainty in the retrieved value due to measurement noise, and $\mathbf{A} = \partial \hat{x} / \partial x_t$ is the AK matrix (Rodgers, 2000) calculated according to Eq. (2).

$$\mathbf{A} = \left(\mathbf{K}^T \mathbf{S}_y^{-1} \mathbf{K} + \mathbf{S}_a^{-1}\right)^{-1} \mathbf{K}^T \mathbf{S}_y^{-1} \mathbf{K} \tag{2}$$

In Eq. (2), $\mathbf{K}$ is the Jacobian matrix of the forward model, the superscript T represents the transpose operator, $\mathbf{S}_y$ is the CM of the observations and $\mathbf{S}_a$ is the CM of the a priori profile. The retrieval error $\boldsymbol{\delta}$ is calculated applying the gain matrix $\mathbf{G}$ (Rodgers,

2000) to an error $\varepsilon$ on the observations randomly taken from a Gaussian distribution with average equal to zero and CM given by $\mathbf{S}_y$:

$$\boldsymbol{\delta} = \mathbf{G}\boldsymbol{\varepsilon} = \left(\mathbf{K}^T \mathbf{S}_y^{-1} \mathbf{K} + \mathbf{S}_a^{-1}\right)^{-1} \mathbf{K}^T \mathbf{S}_y^{-1} \boldsymbol{\varepsilon} \qquad (3)$$

The CM $\mathbf{S}$ associated with the retrieval error $\boldsymbol{\delta}$ (introduced in Eq. (3)) is given by Eq. (4) (Rodgers, 2000):


$$\mathbf{S} = \langle \boldsymbol{\delta}\boldsymbol{\delta}^T \rangle = \left(\mathbf{K}^T \mathbf{S}_y^{-1} \mathbf{K} + \mathbf{S}_a^{-1}\right)^{-1} \mathbf{K}^T \mathbf{S}_y^{-1} \mathbf{K} \left(\mathbf{K}^T \mathbf{S}_y^{-1} \mathbf{K} + \mathbf{S}_a^{-1}\right)^{-1} \qquad (4)$$

The CM $\mathbf{S}_{total}$ associated with the total error $\boldsymbol{\delta}_{total}$ (that is the difference between the simulated and the true profiles, equal to the random $\boldsymbol{\delta}$ plus the so-called smoothing error, caused by the limited vertical resolution of the measurement; see Eq. (7)), is given by Eq. (5) (Rodgers, 2000):


$$\mathbf{S}_{total} = \langle \boldsymbol{\delta}_{total}\boldsymbol{\delta}_{total}^T \rangle = \left(\mathbf{K}^T \mathbf{S}_y^{-1} \mathbf{K} + \mathbf{S}_a^{-1}\right)^{-1} \qquad (5)$$

It should be noted that through the term $\boldsymbol{\delta}$, it is possible to simulate additional error components with respect to the random one considered in this study, and this fact adds flexibility to the simulation method.

In this study, we use the above formulation to simulate ozone profiles in two spectral bands (UV1 and TIR) for both GEO and
LEO, after considering the instrument specifications and accounting for the differences in the two spectral bands. In particular, considering a fixed geo-location, true profile and a priori information, we obtain the L2 products of the different instruments by choice of $\mathbf{K}$ and $\mathbf{S}_y$, that have been synthetized using the technical requirements of the considered platforms and their foreseen performances.

## 2.3. L2 Product Technical Specifications

In the context of Sentinel missions, the ozone profiles derived from measurements in the UV region will be retrieved from spectral radiances acquired by the UVNS/Sentinel-5 spectrometer onboard Meteorological Operational satellite - Second Generation (MetOp-SG) and by the UVN/Sentinel-4 spectrometer onboard Meteosat Third Generation Sounder (MTG-S). For ozone and other targets observed in the TIR, the atmospheric Sentinel missions will use the operational products of IASI-NG on MetOp-SG and of IRS on MTG.

In the framework of the AURORA project, we simulated ozone products from the instruments mentioned above, by using the information available at the beginning of 2016 (ESA 2012a, ESA 2012b, EUMETSAT 2010, Crevoisier et al. 2014. We applied to these specifications some simplifications: for example, we considered only UV1 band (neglecting UV2, 300-320 nm), we considered simplified spatial sampling (spacing between pixels and shape) and in some cases a different signal-to-noise ratio (GEO-TIR L2 type, see Table 1). Consequently, the dataset of simulated L2 products is not exactly in line with the
specifications currently foreseen for the instruments of interest. Table 1 reports some of the more relevant characteristics of the simulated measurements. It is worth to note that when an instrumental parameter has both a Goal value (the value in case the instrument performs at its best) and a Threshold value (the value that we expect to reach anyhow), the latter is used for the simulation.

A more detailed description of the instrumental and observational features goes beyond the scope of this article. All the relevant
information was reported in the Technical Note on L2 Data Simulations (AURORA 2017) and can also be found in (Cortesi et al., 2018). In the next sections of the paper, we do not directly refer to the reference instruments names, but we use an alternative nomenclature; in particular we refer to UVNS/MetOp-SG as LEO-UV1, to UVN/MTG as GEO-UV1, to IASI-

NG/MetOp-SG as LEO-TIR and IR/MTG as GEO-TIR. Since we simulated instruments with characteristics that differ from the real specifications, we think it is appropriate to use an independent terminology also to avoid misunderstandings.

**2.4. The CDF method**

In this section, we briefly recall the formulas of the CDF method (Ceccherini et al., 2015). We assume to have $N$ independent simultaneous measurements of the vertical profile of an atmospheric species that refer to the same geo-location. Performing the retrieval of the $N$ measurements, we obtain $N$ vectors $\hat{x}_i$ ($i$=1, 2, …, $N$) providing independent estimates of the profile, here assumed to be represented on a common vertical grid. Using as inputs these $N$ measurements, the CDF produces as output a
single product characterized by a profile $x_f$, an AK matrix $\mathbf{A}_f$ and a CM matrix $\mathbf{S}_f$ with the procedure summarized by Eqs. (6). These three quantities are dependent on the input products, $\mathbf{A}_i$, $\mathbf{S}_i$, hereafter referred to as fusing products, and depend on the a priori information ($x_a$, $\mathbf{S}_a$) used as a constraint for the fused product.

| | |
|---|---|
| $\alpha_i = \hat{x}_i - (\mathbf{I} - \mathbf{A}_i)x_{ai} = \mathbf{A}_i x_t + \delta_i + \mathbf{A}_i \delta_{coinc,i}$ | (6a) |
| $\tilde{S}_i = \mathbf{S}_i + \mathbf{A}_i \mathbf{S}_{coinc,i} \mathbf{A}_i^T$ | (6b) |
| $x_f = \left( \sum_{i=1}^{N} \mathbf{A}_i^T \tilde{\mathbf{S}}_i^{-1} \mathbf{A}_i + \mathbf{S}_a^{-1} \right)^{-1} \left( \sum_{i=1}^{N} \mathbf{A}_i^T \tilde{\mathbf{S}}_i^{-1} \alpha_i + \mathbf{S}_a^{-1} x_a \right)$ | (6c) |
| $\mathbf{A}_f = \left( \sum_{i=1}^{N} \mathbf{A}_i^T \tilde{\mathbf{S}}_i^{-1} \mathbf{A}_i + \mathbf{S}_a^{-1} \right)^{-1} \sum_{i=1}^{N} \mathbf{A}_i^T \tilde{\mathbf{S}}_i^{-1} \mathbf{A}_i$ | (6d) |
| $\mathbf{S}_f = \left( \sum_{i=1}^{N} \mathbf{A}_i^T \tilde{\mathbf{S}}_i^{-1} \mathbf{A}_i + \mathbf{S}_a^{-1} \right)^{-1} \sum_{i=1}^{N} \mathbf{A}_i^T \tilde{\mathbf{S}}_i^{-1} \mathbf{A}_i \left( \sum_{i=1}^{N} \mathbf{A}_i^T \tilde{\mathbf{S}}_i^{-1} \mathbf{A}_i + \mathbf{S}_a^{-1} \right)^{-1}$ | (6e) |
| $\mathbf{S}_{f\,total} = \left( \sum_{i=1}^{N} \mathbf{A}_i^T \tilde{\mathbf{S}}_i^{-1} \mathbf{A}_i + \mathbf{S}_a^{-1} \right)^{-1}$ | (6f) |

Concerning the profile and the error, we can consider the CDF as a "smart average" in which the a priori information is removed from the L2 profiles and CMs before they are put together in the average. The total error of the L2 product without a priori is higher than the original one, and the effect of the average only partially compensates this error increase. Consequently, even if the total error of the fused product is generally lower than the one of the single L2 fusing product, it is in general higher than the error of the average. The behaviour of the AK matrix is less intuitive, and we will thoroughly analyse it in the
presentation of the results.

If the input products are not coincident in time and space, the CDF introduces a coincidence error characterized by a CM $\mathbf{S}_{coinc}$. In this work, we calculated the diagonal elements of $\mathbf{S}_{coinc}$ as the square of the 5% of the a priori profile $x_a$, where we choose this value considering the size of the coincidence grid cells used in this study. We calculate the off-diagonal elements of $\mathbf{S}_{coinc}$, applying an exponential decay with a correlation length of 6 km (Ceccherini et al. 2018). The 5% choice matured in a heuristic
way by varying the percentage value and observing the quality of the fused product in some single reference cells and by looking at the entire dataset in representations similar to figures 6 and 7.

The dynamical choice of $\mathbf{S}_{coinc}$ is presented in (Ceccherini et al. 2019). In particular the a priori error (coincident with the climatological variability) is used as the reference for the diagonal elements and a fixed exponential decay is applied too. However, the multiplicative factor is calculated by imposing that the cost function of the retrieval is equal to its expected value.

That study, which is based on simulated products similar to the ones of this work, shows that even if the coincidence error is strictly needed for the correct behaviour of the CDF product, this is not strongly dependant by its exact amount until it is smaller than the errors of the individual L2 products.

The formulas of Eqs. (6) refer to the case of measurements made on the same vertical grid. In general, also an interpolation error may be needed considering that the retrievals of the products to be fused can be defined on different vertical grids. In

(Ceccherini et al. 2018) the general expressions of CDF in the case of the fusion of products characterized by different vertical grids are presented and discussed together with the expression of the interpolation error that depends on the involved grids and the AK matrices of the fusing products. However, since the interpolation error does not apply to the present study (we simulated all the L2 products on the same vertical grid), it has not been considered in Eqs. (6) and in the following discussion.

### 2.5. Arithmetical average and biases

Before proceeding, it is necessary to clarify why the arithmetic average of the profiles cannot be considered as a good option to represent a set of products retrieved with optimal estimation techniques.

To do this, we consider $N$ coincident L2 measurements ($i=1,.., N$) referring to the same true profile, the same AK matrix and the same CM but having different (noise) errors $\delta_i$ randomly generated according to Eq. (3). The total error expression for the $i$-$th$ measurement is given in Eq. (7) that can be easily derived from Eq. (1).


$$\delta_{i,\text{total}} = \hat{x}_i - x_t = (\mathbf{I} - \mathbf{A}_i)(x_a - x_t) + \delta_i \tag{7}$$

Considering that the individual measurements are co-located in space and time, thus they refer to the same truth, the same a priori profile and the same AK matrix $\mathbf{A}$, the mean total error is equal to:

$$\langle\delta_{i,\text{total}}\rangle = \langle\hat{x}_i\rangle - x_t = (\mathbf{I} - \mathbf{A})(x_a - x_t) + \frac{1}{N}\sum_{i=1}^{N}\delta_i \tag{8}$$


It follows that the averaging process reduces the random component of the total error, but does not reduce the bias due to the a priori information. This bias is equal to the term $(\mathbf{I} - \mathbf{A})(x_a - x_t)$ of Eq. (8), which therefore becomes a dominant component as the number N increases. The existence of this bias is one of the reasons why the arithmetic average cannot be considered as a reference algorithm to collect the information of several products into one. Further reasons concern the choice

of a suitable AK matrix to be assigned to the average (see also von Clarmann. and Glatthor 2019) and the management of possible coincidence and interpolation errors. An explicative comparison of the application of CDF and standard averages in the case of 1000 coincident L2 products is reported in the supplementary material.

## 3. Results and discussion

### 3.1. Fusion in realistic spatial and temporal resolution conditions: the L2 Datasets

To analyse the behaviour of CDF in realistic spatial and temporal resolution conditions, we consider four sets of measurements. These measurements correspond to the cloud-free observations that were possible between 9:00 am and 10:00 am on the 1[st] April 2012. Table 2 lists the L2 product types, namely GEO-TIR, GEO-UV1, LEO-TIR and LEO-UV1, used in this article. The L2 datasets have been generated according to the equations (1)-(5) described in paragraph 2.2. The details of the simulation process can be explored in the technical note (AURORA 2017) considering that here we simulated all the pixels corresponding

to a clear sky line of sight in the atmospheric scenario, without applying any additional selection criteria. In the AURORA project main workstream, we considered four months of data; however, we simulated only a subset of the clear-sky pixels to

reduce the computational cost of the simulations (Tirelli et al. 2020). In this AURORA side-study, we consider one hour of data, and we simulate all the clear-sky pixels without additional filters, choosing the orbits so that GEO-LEO coincidences occur. The left panel of Figure 5 indirectly represents the spatial distribution of the products simulated for this study.


### 3.2. Single grid-box analysis (0.5°x0.625°)

We consider the case of a single grid-box (Figure 1). In the selected grid-box, 118 measurements were available (55 of GEO-TIR, 55 of GEO-UV1, 8 of LEO-TIR, no LEO-UV1). The cell has the size of 0.5 degrees in latitude and 0.625 degrees in longitude, centred on the Egina Island in the Aegean Sea. The cell size has been chosen to be comparable with the assimilation

grid used in the AURORA project. We assign the geo-location of the fused product to be the barycentre of the horizontal coordinates of the L2 measurements in the grid-box. In this particular case, since the horizontal distribution of the 118 L2 profiles is quite homogeneous, the barycentre is placed at the centre of the grid-cell.

Figure 2 shows with green lines the absolute (left panel) and relative (right panel) differences between each L2 profile and the corresponding true profile, with a red line the difference between the fused profile and the mean truth (computed as the average

of the 118 true profiles), with a black dash-dotted line the average of the estimated standard deviation of the total error of the individual L2 measurements $\sigma_{total}$, and with a red dash-dotted line the estimated standard deviation of the total error of the fused profile $\sigma_{f\,total}$. The last two quantities have been calculated as the square root of the diagonals of the $\mathbf{S}_{total}$ and $\mathbf{S}_{ftotal}$ CMs given by Eqs. (5) and (6), respectively. Figure 2 shows that the fused product is in better agreement with its truth than the individual profiles with their own, and presents a smaller estimated total error than the individual L2 products. In particular,

the right panel allows seeing in detail the performances of CDF in the tropospheric region.

The retrieved profile representation is always a compromise between the amplitude of the errors and the vertical resolution. The latter can be quantified by the AKs, which ideally would be equal to the identity matrix in the case of a profile that has a vertical resolution equal to that defined by the sampling grid. Diagonal elements with values smaller than 1 correspond to a loss of vertical resolution. In the left panel of Figure 3, we compare the diagonal elements of the AKs of the L2 products with

the AK diagonal of the fused product. Here we have also computed the number of Degrees Of Freedom (DOFs), given by the sum of the diagonal elements of the AK matrix (Rodgers, 2000), for both L2 and fused products, and reported the values in the text box of the left panel. Note that the number of DOFs of the fused product is about twice the number of DOFs of the best L2 one. In the right panel of Figure 3, we compare the vertical resolution profiles of L2 and FUS products. We calculate the vertical resolution starting from AK matrices according to the Full-Width-Half-Maximum (FWHM) approach (Rodgers,

2000), and in particular with the algorithm defined in (Ridolfi and Sgheri, 2009).

From the comparison of the left and the right panel of Figure 3, it can be noted that the increase of the AK matrix diagonal values of FUS product, and consequently the increase of the number of DOFs, implies an improved vertical resolution only in a subset of the vertical levels. To better understand the effect of the fusion on the AK matrices, it is useful to analyse the behaviour of their rows. In Figure 4, two rows are represented, one that refers to the troposphere (left panel, 6 km) one to the

middle stratosphere (right panel, 39 km), where the reference altitude is the one corresponding to the diagonal value of the row. The value of the vertical resolution at the considered altitude is reported in the legend (the minimum of vertical resolution at the considered vertical level for each type of L2 product), and the diagonal value of each row is evidenced in the graphs with cross (L2) and dot markers (FUS). At lower altitudes (left panel), the DOFs increase can be attributed to three distinct phenomena. The first is the constriction of the main FUS AK lobe and the consequent improvement (of more than 30%) of the

vertical resolution with respect to L2 products. The second phenomenon is linked to the fact that while for the FUS product the maximum value of the AK row corresponds to its diagonal element, for the L2 products these maxima are shifted with respect to the reference altitude of the rows. The last phenomenon is a stronger contribution of the (simulated) measurements with respect to the a priori in the FUS product, where the latter effect can be evidenced considering the sum of all the elements

of the rows that assume 0.913 as the maximum value for the L2 products and 0.956 for the FUS product. In this particular
case, all these three effects go in such a direction that can be considered as benefits of CDF application. The results at higher
altitudes (39 km, right panel) are primarily influenced by the shape of the AK rows that exhibit large secondary lobes that
degrade the vertical resolution.

### 3.3. Statistical analysis for a large domain

While the analysis of the previous paragraph focuses on a particular grid-box, here an analysis of the CDF behaviour is
presented, referring to all the 1939 fusion grid-boxes in which more than one of the 79781 L2 simulated products, considered
in Table 2, is placed. The fused products can be classified depending on the types of L2 measurements falling inside the
coincidence grid cell. Since GEO-TIR and GEO-UV1 products are in perfect coincidence and LEO-UV1 products have a
horizontal spacing larger than the cell size, only six fused product types (FUS type), listed in Table 3, effectively occur. In this
table, the FUS type and its description are reported together with the following complementary data:

- Ncells: the number of grid-boxes characterized by the considered FUS type.
- <NL2>: the mean number of individual L2 fusing profiles per grid-box.
- Max NL2: the maximum number of individual L2 fusing products per grid-box.

The left-hand side panel of Figure 5 shows the geographical distribution of the FUS products. Different colours have been
used to classify the fused data according to their provenance type. The irregular geographical coverage is due to the realistic
distribution of the cloud-free measurements. The histogram in the right-hand side panel of Figure 5 shows the number of cells
that contain a given number of measurements, divided in different colours depending on the FUS type. The FUS cells, in which
only LEO products fall, are characterized by a small number of L2 measurements, while when GEO products are present,
many L2 measurements can be present.

With the selected grid-box size and the multitude of different products that are present in each cell, the question is which
product can be used in alternative to the fusion process in those operations in which a single product is requested in each grid-
box. Since the averaging process is affected by a large bias error, a viable alternative is the use of the best fusing product
present in the cell, and we want to compare the CDF result with this product. This comparison is the so-called Synergy Factor
(SF), introduced by Aires et al. (2012). Although Aires introduces SF only for errors (Eq. (11)), we extend his definition also
for other quantities because they constitute a useful tool to synthetically represent the performances of fusion algorithms.
The *SF DOF*, defined by Eq. (9), is the ratio between the number of DOFs of the FUS product, and the maximum number of
DOFs of L2 fusing products. In this equation, the index $l$ enumerates the vertical levels and the index $i$ enumerates the L2
products fused in each grid-box.

$$SFDOF = \frac{\sum_l \mathbf{A}_{f,ll}}{\max_{i \in L2} \sum_l \mathbf{A}_{ll}}$$ (9)

When *SF DOF* is larger than 1.0, the FUS product carries more information than the individual L2 measurements. Figure 6
shows that the *SF DOF* computed for all the fused products (and plotted as a function of the number of L2 profiles in each
grid-box) is always larger than 1.0. It is also worth noticing that *SF DOF* increases approximately linearly with the logarithm
of the number of fusing products, although the proportionality depends on the FUS type. The two different clusters of red
symbols (GEO:TIR+UV1) are caused by the different latitude bands in which these products are distributed (see also left panel
of Figure 5). It is important to underline that the improvement in vertical resolution is the most demanding requirement in
remote sensing observations and, considering the significant gain obtained relative to the single product selection, is the most
important feature of fused products.

While *SF DOF* is a scalar quantity, both **SF AK** and **SF ERR**, defined by Eqs. (10) and (11), are vertical profiles of pure numbers. **SF AK** represents an expansion on the vertical dimension of *SF DOF* and, in particular, is calculated, level by level,

as the ratio between the diagonal elements of the AK matrix of the FUS product and the maximum of the corresponding elements of the AK matrices of the fusing L2 measurements.

A value of **SF AK** larger than 1.0 at a specific vertical level (indicated by the index *l*) means that, at that level, the diagonal value of the AK matrix of the FUS product has a larger value than that of all the individual products. As we have seen commenting Figure 3 and Figure 4, the increase of the AK diagonal values at a specific level can happen for different reasons,

but all of them can be considered as an improvement in the product quality.

$$SFAK_l = \frac{\mathbf{A}_{f,ll}}{\max_{i \in L2}\mathbf{A}_{i,ll}} \qquad (10)$$

The **SF ERR** (Eq. (11)) at a given level is the ratio between the minimum total error of the L2 measurements that have been fused and the total error of the FUS product. A value of **SF ERR** larger than 1.0 means that, at a specific level, the error of the

FUS product is smaller than that of all the individual products.

$$SFERR_l = \frac{\min_{i \in L2}\sigma_{total,i,l}}{\sigma_{ftotal,l}} \qquad (11)$$

The SFs defined by Eqs. (10) and (11) provide a conservative comparison because the fused product is compared with the L2 product that at that level has the largest diagonal value in its AK matrix and with the one that has the smallest total error at the

same level (generally, these are two distinct L2 products).

Figure 7 shows the **SF AK** (left panel) and **SF ERR** (right panel) profiles for the 1939 FUS products considered in Table 3. We have used different colours to denote the provenance of the L2 data contributing to the fused products and different symbol size to infer the number of L2 fusing measurements (the larger the symbol size, the larger the number of L2 fusing profiles). The significant improvement obtained with the fused products is confirmed by Figure 6. In Figure 7, considering symbols of

the same colour, the size (N) increases moving horizontally in the graph (same vertical level) from left to right (SF increasing). This fact denotes that, for each FUS type, SF increases with *N*. This is not in contradiction with the fact that symbols with different colours (FUS types) and different sizes (*N*) can share the same position (SF, vertical level) on the graph. Some **SF AK** values, both in the troposphere and in the middle-upper atmosphere are smaller than one; in the troposphere, this happens in 20 cells out of 1939 while in the middle-upper atmosphere this happens in almost 500 cells for two possible and sometimes

simultaneous circumstances. The first one happens when the introduction of the coincidence error provokes a sensible degradation of the quality of the FUS AK matrix. The second circumstance happens, for example, when one of the L2 products is characterized by a vertical resolution that is much better than all the other fusing products and, in particular, the peaks of their AK rows tends to not coincide with the nominal vertical level of the row itself.

### 3.4. Statistical analysis on a coarse horizontal resolution

We have seen that starting from 79781 L2 measurements (Table 2), when a coincidence grid-box with size 0.5°x0.625° is used, the number of fused profiles is 1939 (Table 3), with a reduction of the data volume of more than a factor 40.

Table 4 provides a summary of the number of fused profiles and the provenance of the L2 profiles that contribute to them for a fusion grid resolution of 1°x1°. In this case, the total number of FUS products is 775, with a reduction of the data volume of

more than a factor 100.

The Synergy factors *SF DOF*, **SF AK** and **SF ERR**, also in this case have been considered, and the figures (similar to Figure 6 and Figure 7) are reported in the supplementary material. In summary, the greater number of fusing observations in each fusion cell produces a further improvement for both the vertical resolution and the total error. This observation confirms that the CDF method can be used with a wide range of grid-box size and data compression and the quality of the products generally

improves with larger cells. An upper limit to the grid-box size is caused by the coincidence error amplitude, which increases with the geographical variability, degrading the quality of the fused product. The study of this aspect will be of crucial importance if the CDF will be applied to species with greater spatial-temporal variability than ozone or in any case, to very large spatial-temporal domains.

## 4. Conclusions

This paper presents a feasibility study of the CDF technique applied to L2 products simulated according to the characteristics of the atmospheric Sentinel missions. Despite the approximations that characterize the simulated L2 products (technical specifications not exactly in line with the ones of the atmospherics Sentinels and no systematic errors added) this analysis allows to evaluate the performances of the CDF algorithm in a realistic scenario

In particular, we show the application of CDF to a single cell with a size of 0.5 degrees in latitude and 0.625 degrees in

longitude in which more than 100 L2 products are fused. Results show that the fused product is characterized by higher information content, smaller errors and smaller residuals (i.e., smaller anomalies from the true profiles) compared to individual L2 products. The information content being, with its improvement of the vertical resolution, the most important achievement. This analysis is then extended to a larger domain consisting of 79781 L2 products subdivided into 1939 grid boxes with 0.5°x0.625° size. In this case, the comparison of L2 products and CDF output are carried on in terms of synergy factors. This

analysis shows that the CDF can be applied to a wide range of situations and that the benefits of the fusion strongly depend on the number of measurements that are fused and on their characteristics. It is also shown that CDF can be run customizing grid resolutions, e.g. to match the resolution requirements of the process that will ingest the products, with full exploitation of all the available measurements.

As the fused products are traced back to a regular, fixed horizontal grid and, as shown here, are not affected by the bias

introduced by the a priori information, they can be considered as a new type of Level 3 products with improved quality (reduced bias) and the same characteristics (AK included) with respect to L2 products, even if further analysis is needed, especially for what concerns the coincidence error to be applied to fuse data on large spatial-temporal domains.

**Data availability**

The data of the simulations presented in the paper are available from the authors upon request.

MERRA-2 data (atmospheric scenario) are available at MDISC (https://disc.gsfc.nasa.gov), managed by the NASA Goddard Earth Sciences (GES) Data and Information Services Center (DISC).

The ML climatology (McPeters and Labow, 2012) is available online from the Goddard anonymous ftp account: ftp://toms.gsfc.nasa.gov.

**Author contributions (according to CRediT https://casrai.org/credit/)**

N. Zoppetti: Conceptualization, Methodology, Software, Writing – Original Draft, Writing – Review & Editing, Investigation, Data curation, Visualization S. Ceccherini: Conceptualization, Methodology, Investigation, Writing – Review & Editing B. Carli: Conceptualization, Methodology Writing – Review & Editing, Supervision S. Del Bianco: Investigation, Data curation, Project Administration M. Gai: Investigation, Data curation C. Tirelli: Investigation, Data curation, Project Administration F.

Barbara: Resources R. Dragani: Investigation, Data curation, Writing – Review & Editing A. Arola: Investigation, Data
curation J. Kujanpää: Investigation, Data curation R. Van Der A: Investigation, Data curation U. Cortesi: Project
Administration, Supervision, Writing – review & editing.

**Competing interests.**

The authors declare that they have no conflict of interest.

**Acknowledgments**

The results presented in this paper arise from research activities conducted in the framework of the AURORA project
(http://www.aurora-copernicus.eu/) supported by the Horizon 2020 research and innovation programme of the European Union
(Call: H2020-EO-2015; Topic: EO-2-2015) under Grant Agreement N. 687428.

**Financial support.**

This research has been supported by the European Commission, H2020 (AURORA, grant no. 687428).

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

| L2-Type Name | GEO-TIR | GEO-UV1 | LEO-TIR | LEO-UV1 |
|---|---|---|---|---|
| Platform[1] | Meteosat Third Generation Sounder (MTG-S) | | Meteorological Operational satellite - Second Generation (MetOp-SG) | |
| Reference instrument[1] | Infrared Sounder (IRS) | UV-VIS-NIR Sentinel-4 spectrometer (UVN) | Infrared Atmospheric Sounding Interferometer – New Generation (IASI-NG) | UV-VIS-NIR-SWIR Sentinel-5 spectrometer (UVNS) |
| Retrieval spectral range | 1030-1080 cm$^{-1}$ | 305-320 nm | 1030-1080 cm$^{-1}$ | 270-300 nm |
| Spectral resolution | 0.625 cm$^{-1}$ (apodized IRSF) | 0.5 nm | 0.25 cm$^{-1}$ (apodized IRSF) | 1.0 nm |
| Spectral sampling ratio | 1 | 3 | 2 | 3 |
| Noise Equivalent Brightness Temperature Scenario (according to Crevoisier et al. 2014) | IRS1b | | IRS2b | |
| Signal to Noise Ratio Radiance | | 160@305 nm (T) 320@310 nm (T) 630@315 nm (T) 900@320 nm (T,G) | | 100 @270 nm (T,G) |

| Field of View | $15\times15$ km² (T) $5\times5$ km² (G) | <= 8 km at 45°N and longitude of the satellite (0°E) | $12\times12$ km² (T) $5\times5$ km² (G) | $50\times50$ km² (T) $15\times15$ km² (G) |
|---|---|---|---|---|
| Pixel size used in the simulation (assumed square @ nadir) | 8 km | | 12 km | 45 km |

**Table 1: instrument characterization relevant for the simulation process. Goal (G) and Threshold (T) values correspond, respectively, to estimates of the parameters in case the instrument performs at its best and to limit values that we expect to reach anyhow. AURORA will be using Threshold values for the generation of simulated data.**
[1] The table entries in these rows are not meant to suggest that the specifications listed in this table refer directly to these instruments. Instead, the listed instruments are examples of instruments actually planned, with similar, however, slightly different specifications.

| L2 Type | Number of simulated measurements | Minimal distance between measurements across x along track [km] |
|---|---|---|
| GEO-TIR | 35594 | 5.7 x 7.4 |
| GEO-UV1 | 35594 | |
| LEO-TIR | 8023 | 12.2 x 12.3 |
| LEO-UV1 | 570 | 46.2 x 46.7 |
| TOTAL | 79781 | |

**Table 2: Number and horizontal resolution of the simulated measurements. For GEO platform across-track is South-North direction, and along-track is East-West direction.**


| FUS Type | Description | $N_{cells}$ | $<NL_2>$ | max NL2 |
|---|---|---|---|---|
| GEO:TIR+UV1 | Two or more GEO pixels, no LEO pixels. | 908 | 29.3 | 160 |
| GEO:TIR+UV1_LEO:TIR+UV1 | Two or more GEO pixels, one or more LEO-TIR pixel, one or more LEO-UV1 pixel. | 245 | 114.7 | 163 |
| GEO:TIR+UV1_LEO:TIR | Two or more GEO pixels, one or more LEO-TIR pixel, no LEO-UV1 pixels. | 299 | 69.4 | 165 |
| GEO:TIR+UV1_LEO:UV1 | Two or more GEO pixels, one or more LEO-UV1 pixel, no LEO-TIR pixels. | 2 | 20 | 37 |
| LEO:TIR+UV1 | No GEO pixels, one or more LEO-TIR pixels, one or more LEO-UV1 pixels. | 247 | 11.1 | 24 |
| LEO:TIR | No GEO pixels, two or more LEO-TIR pixels, no LEO-UV1 pixels. | 238 | 6.2 | 14 |
| TOTAL | | 1939 | 41.1 | 165 |

**Table 3: types and characteristics of the fused product when a coincidence grid cell size of 0.5ºx0.625º is used. Ncells is the number of grid-boxes characterized by the considered FUS type; <NL2> is the mean number of individual L2 fusing profiles per grid-box and Max NL2is the maximum number of individual L2 fusing products per grid-box.**

| FUS Type | Description | $N_{cells}$ | $<NL_2>$ | max NL2 |
|---|---|---|---|---|
| GEO:TIR+UV1 | Two or more GEO pixels, no LEO pixels. | 354 | 73.1 | 420 |
| GEO:TIR+UV1_LEO:TIR+UV1 | Two or more GEO pixels, one or more LEO-TIR pixel, one or more LEO-UV1 pixel. | 140 | 289.4 | 504 |
| GEO:TIR+UV1_LEO:TIR | Two or more GEO pixels, one or more GEO-TIR pixel, no LEO-UV1 pixels. | 79 | 115.4 | 442 |
| GEO:TIR+UV1_LEO:UV1 | Two or more GEO pixels, one or more LEO-UV1pixel, no LEO-TIR pixels. | 0 | 0 | 0 |
| LEO:TIR+UV1 | No GEO pixels, one or more LEO-TIR pixels, one or more LEO-UV1 pixels. | 142 | 26.2 | 71 |
| LEO:TIR | No GEO pixels, two or more LEO-TIR pixels, no LEO-UV1 pixels. | 60 | 8.9 | 26 |
| TOTAL | | 775 | 102.9 | 504 |

**Table 4: Like in Table2 but with a grid-box size of 1ºx1º. Ncells is the number of grid-boxes characterized by the considered FUS type; <NL2> is the mean number of individual L2 fusing profiles per grid-box and Max NL2is the maximum number of individual L2 fusing products per grid-box.**

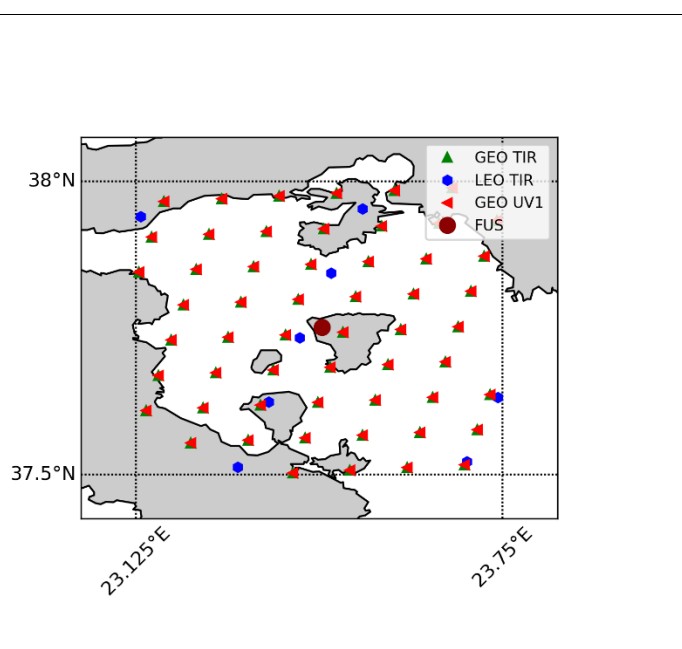

**Figure 1: geographical distribution of the simulated L2 measurements and geo-location of the fused product. The black dash-dotted lines represent the borders of the 0.5ºx0.625º grid cells.**

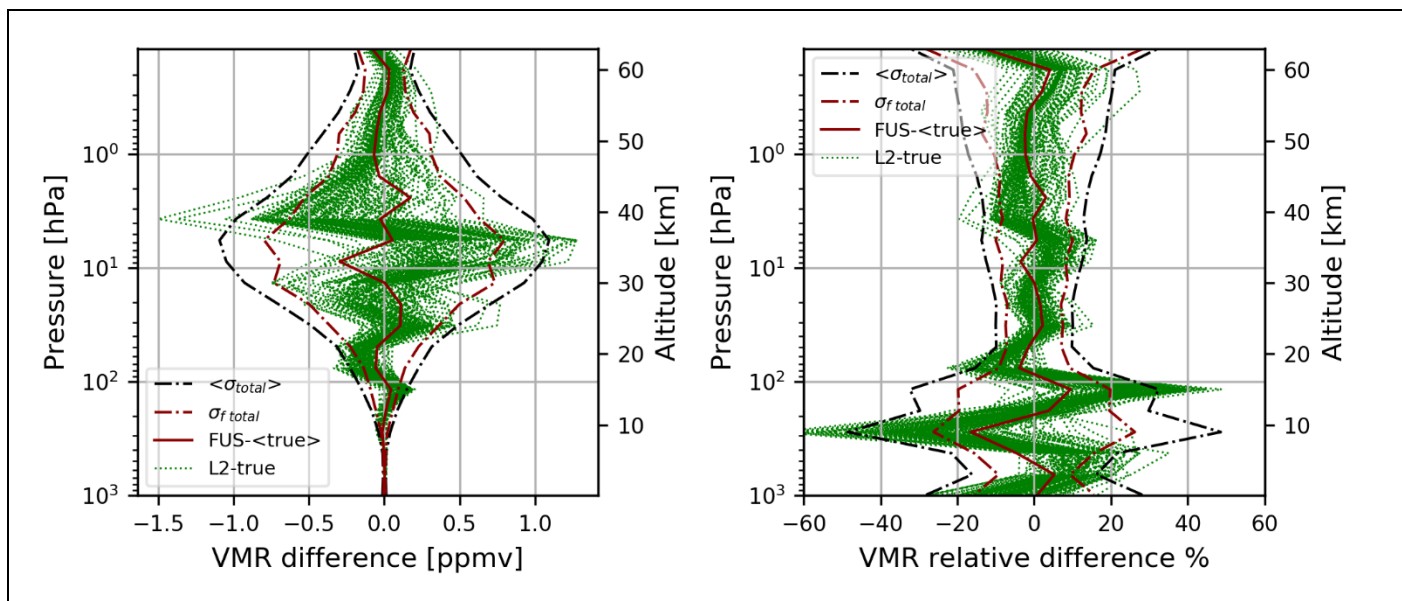

**Figure 2 (Left panel): the absolute differences between L2 profiles and their true profiles (green lines), the absolute difference between the fused profile and the average of the true profiles (dark red continuous line), the average of $\sigma_{total}$ of L2 simulations (black dash-dotted lines), $\sigma_{f\ total}$ (dark red dash-dotted lines). (Right panel): the relative percentage differences between L2 profiles and their true profiles (green lines), the relative percentage difference between the fused profile and the average of the true profiles (dark red continuous line), the average of $\sigma_{total}$ of L2 simulations normalized wrt the true profile and expressed in percentage (black dash-dotted lines), $\sigma_{f\ total}$ normalized wrt the true profile and expressed in percentage (dark red dash-dotted lines).**

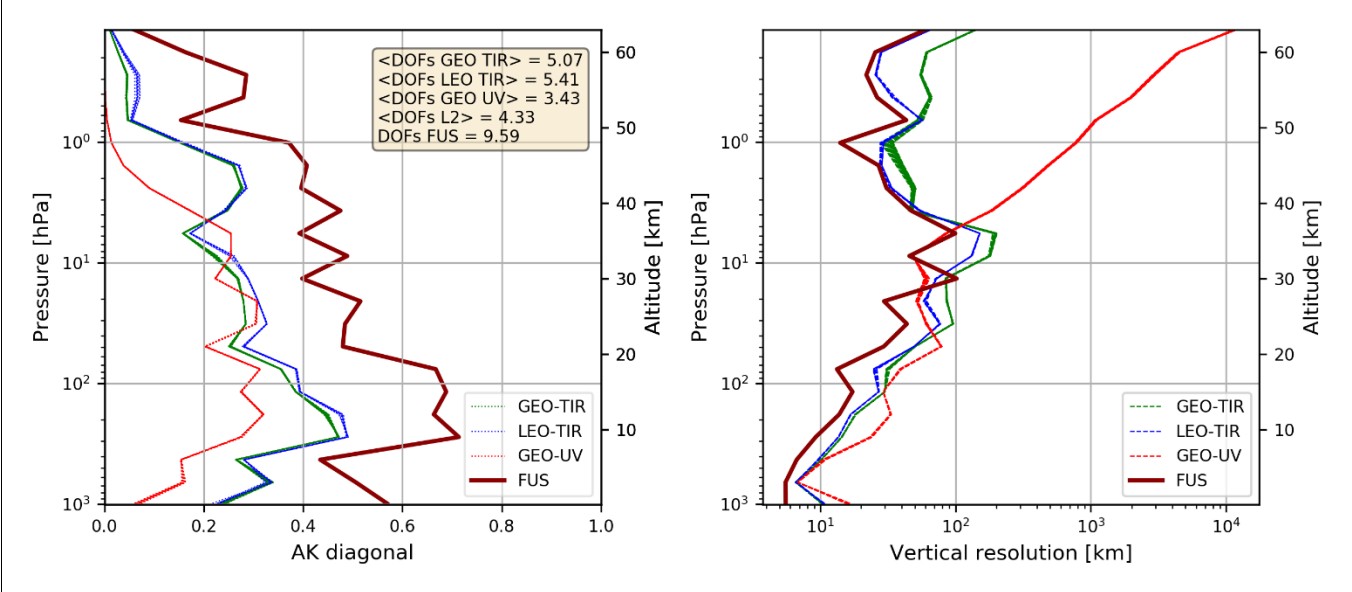

**Figure.3 (Left panel):** AKs diagonals. GEO-TIR products (red lines), LEO-TIR products (blue lines) GEO-UV products (red lines) and FUS product (dark red line). In the text box, the average number of DOFs for each type of L2 product, the average number of DOFs for all L2 products and the number of DOFs of the FUS product are reported. **(Right panel):** Vertical resolution (FWHM) profiles. GEO-TIR products (red lines), LEO-TIR products (blue lines) GEO-UV products (red lines) and FUS product (dark red line). In each panel, while the solid dark red line is a single one, the red and green lines are both 55 overlapped lines, and the blue lines are eight overlapped lines (one for each L2 product).


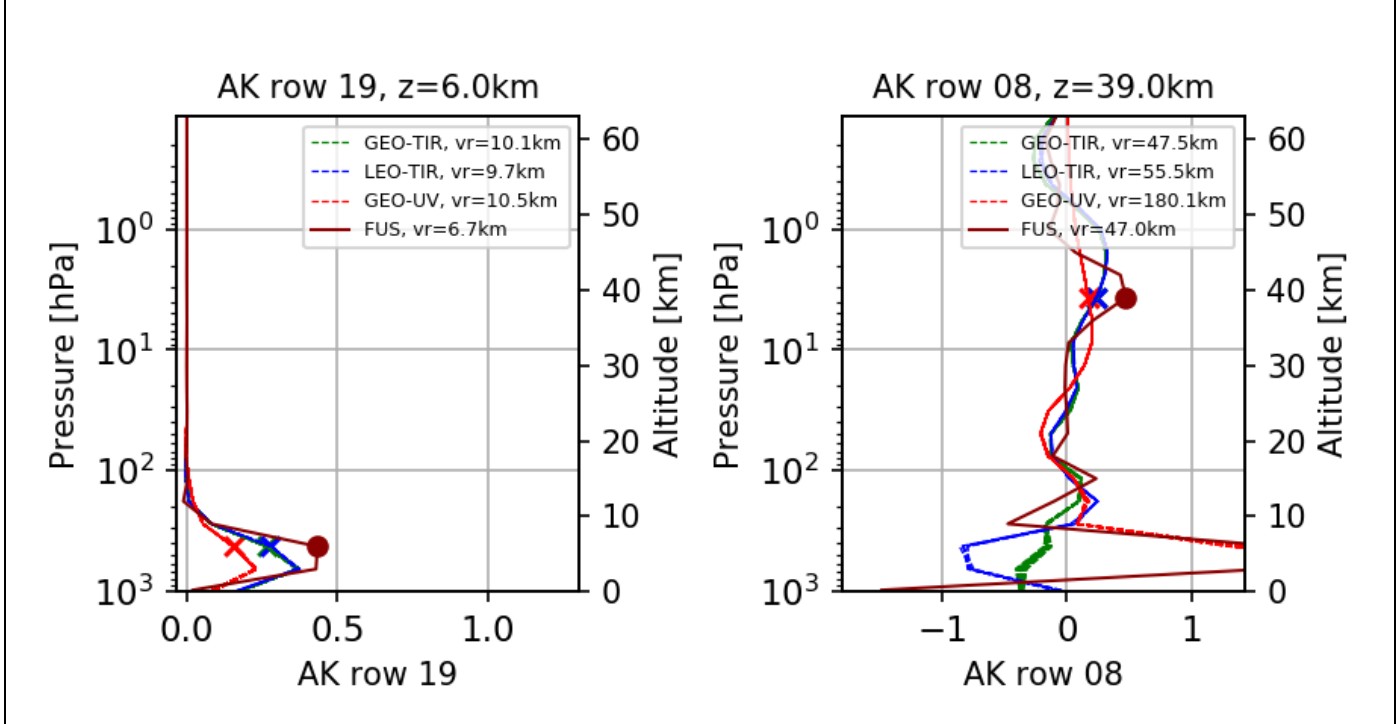

**Figure 4 (Left panel):** Rows of AK matrices at 6 km. **(Right panel):** Rows of AK matrices at 39 km GEO-TIR products (red lines), LEO-TIR products (blue lines) GEO-UV products (red lines) and FUS product (dark red line).

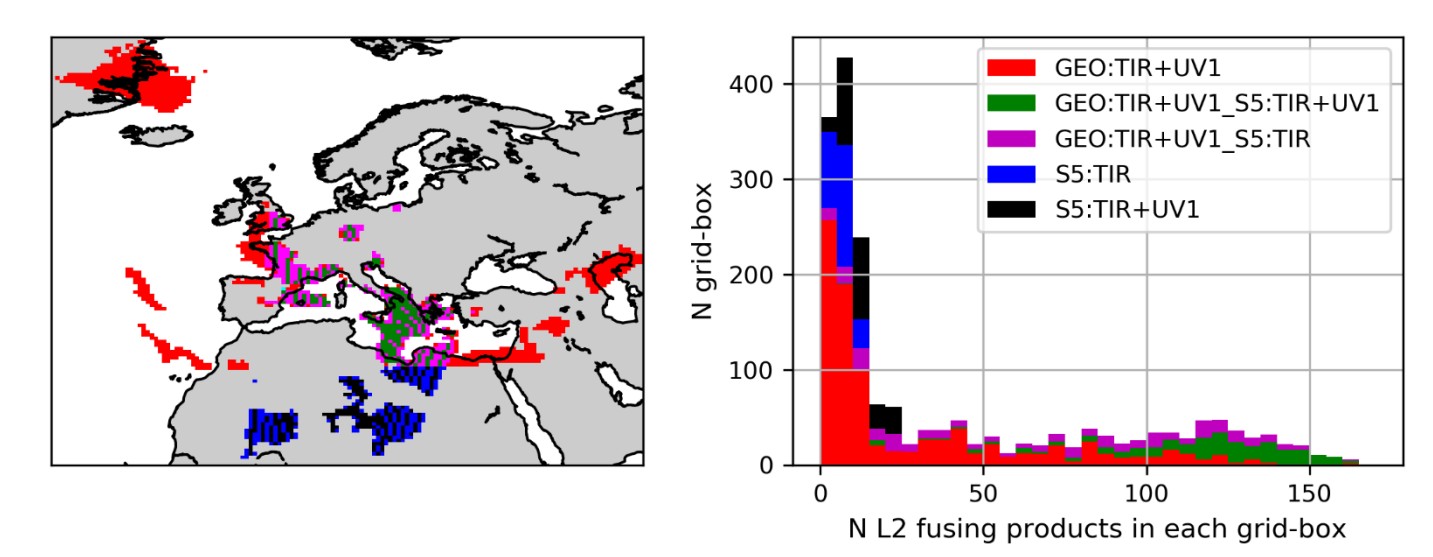

**Figure 5. Left panel: geographical distribution of FUS products differentiated by FUS type where the effect of the lower resolution of LEO-UV1 respect to the other L2 products is the cause of the periodic FUS type transitions in the Mediterranean area. Right panel: histogram of the number of cells with a given number of L2 measurements differentiated by FUS type.**

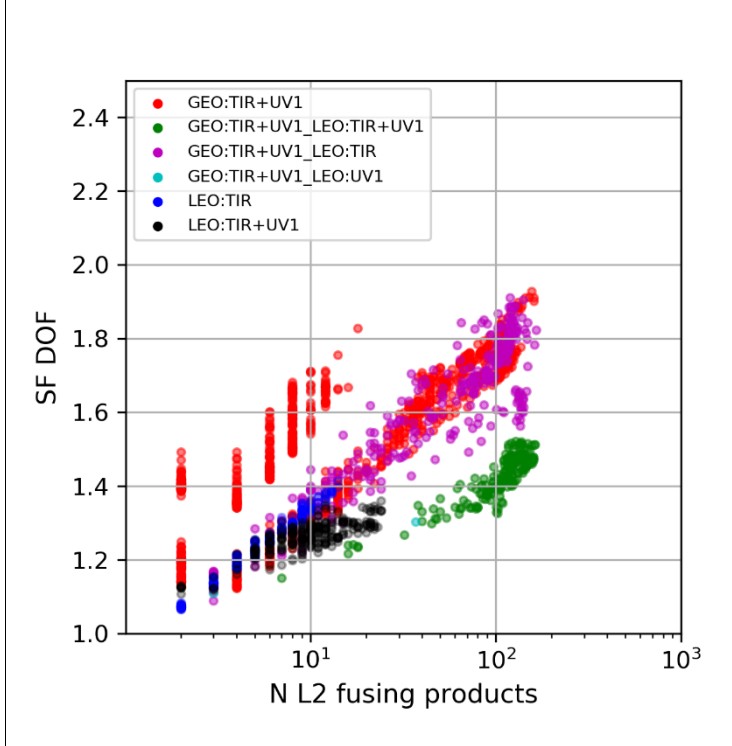

**Figure 6: scatter plot of SF DOF as a function of the number of L2 measurements fused in each coincidence grid cell; different colours represent different FUS types.**

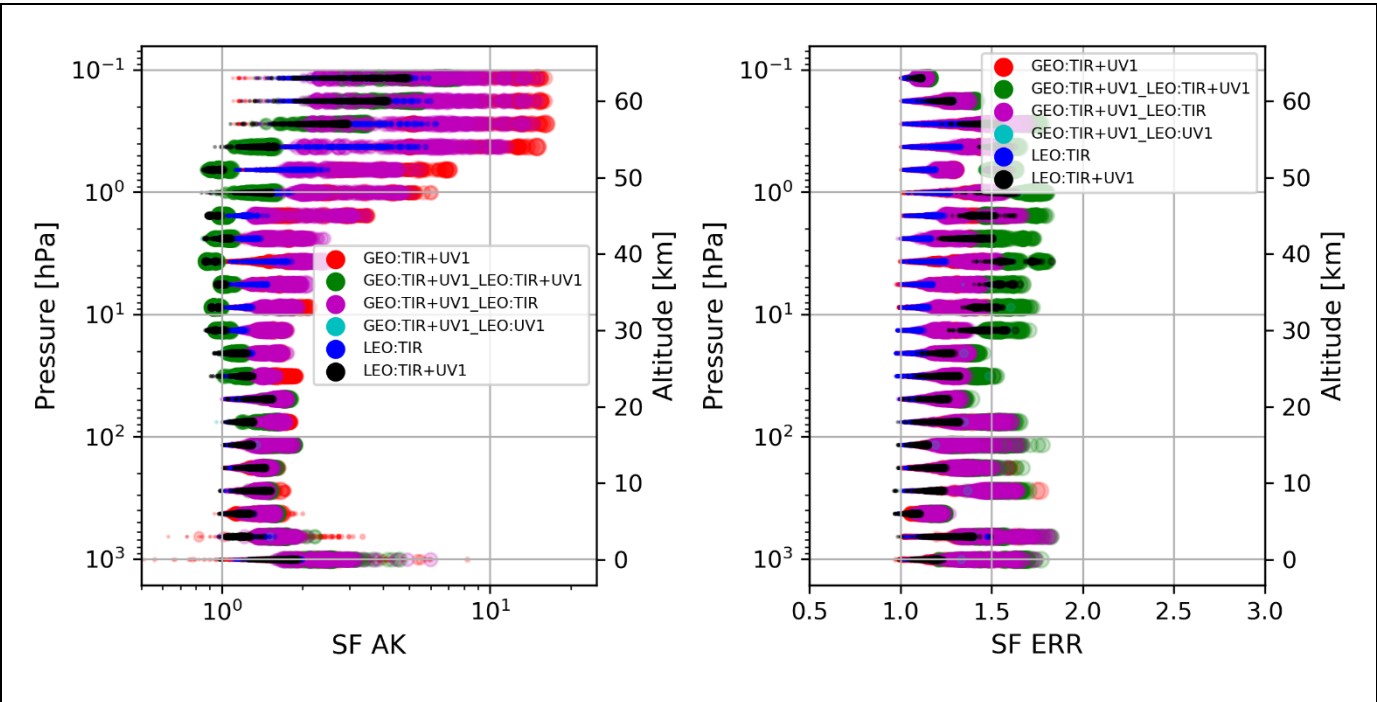

**Figure 7 (Left panel): SF AK versus vertical level. (Right panel): SF ERR versus vertical level. In both panels, different colours of the symbols represent the FUS type, different sizes of the symbols represent the number of measurements that have been fused. The maximum symbol size shown in the legend corresponds to N=160.**
