# Peer review of "Application of the Complete Data Fusion to the ozone profiles measured by geostationary and low earth orbits satellites: a feasibility study"

_Atmospheric Measurement Techniques, 2019_

## Referee Comment (RC1) · Anonymous Referee #1 · 14 Jan 2020

**General comments:**

In this work the authors demonstrate that the Complete Data Fusion method, which they have previously developed themselves, is very well suited for the combination of large amounts of retrieved satellite data without information loss. The demonstration is based on (here simulated) Sentinel-4 and Sentinel-5 data, for which the future use of the Complete Data Fusion method could be of high relevance. Although the importance of the Complete Data Fusion method and its insightful demonstration in this work is acknowledged, it is believed that the authors should address two important issues before the work can be accepted for publication. First, in the "Results and discussion"

section, starting from line 186 and into the conclusions, the authors assume that a DFS increase automatically implies an increased vertical resolution of the retrieval. This is not true, as a DFS change can also be induced by a change of the weight of the measurement (or, equivalently, of the prior information) in the retrieval, as represented by the averaging kernel matrix' row sums, or by a change in the retrieval height offset (vertical shift of the measurement weight). Both are independent of the retrieval's vertical resolution, which e.g. can be determined from a kernel's FWHM. It is very important that the authors consider these alternatives based on a thorough discussion of (fused) averaging kernel matrix behavior and either modify their discussion and conclusions accordingly or demonstrate that only the vertical resolution is impacted by the method. The lack of a thorough averaging kernel discussion also points at the second issue: The presentation of this work appears somewhat sloppy, with important information being vague or even missing, while it certainly deserves comprehensiveness. Several suggestions for improvement are therefore being made in the specific and technical comments below.

Specific comments:

- The averaging of profiles as a simple combination technique is not mentioned in the abstract nor the introduction, while it is being discussed in a separate section eventually. Mentioning a discussion of the Complete Data Fusion method with respect to simple averaging from the beginning would improve readability.

- Stating "the quality of the products improving with larger grid boxes" in the abstract is misleading, as representativeness errors are also expected to increase with the grid box size. The latter is only briefly mentioned in the very last sentence before the conclusions. This important point deserves more discussion (especially regarding the differences in natural variability for different atmospheric molecules) and mentioning in the abstract.

- Line 45: "whenever the user does not need the full spatial and temporal resolution"
sounds misleading, as it seems that the Complete Data Fusion method can also be used to combine measurements from several satellites choosing e.g. the pixel size of one of the contributing instruments?

- Lines 75-77: The specific differences and usages of the ozone climatology and the atmospheric scenario might be far from clear for some readers. It would be very helpful to extend on this and relate this information with the quantities in the equations.

- The sentence (paragraph) on line 120 is very unclear (or trivial) and fully without context.

- Lines 132-133: "the diagonal elements of S\_coinc are calculated as the 5% of profile estimates in the ozone climatology" is not clear at all.

- Lines 135-137: It is agreed that interpolation errors do not apply in this study, but as a demonstration "for a Full Exploitation of Copernicus Atmospheric Sentinel Level 2 Products" (title) the implications of the need for a preceding interpolation for upcoming reality should nevertheless be decently discussed.

- Line 153: The authors could refer to van Clarmann and Glatthor, 2019, for possibly improving their discussion on the averaging kernel matrix of an average product.

- Lines 158-161: This paragraph is way too short for a proper understanding of how the simulated retrievals are spatiotemporally allocated. How are instrument and orbit characteristics taken into account and applied to a past atmospheric scenario?

- Line 168: The calculation of the barycenter should be specified. Why are only horizontal coordinates considered for calculating the location of the fused product? Figure 1 seems to suggest differently.

- Figure 2 and comparisons in supplementary material: Although absolute differences have their benefits, (additionally) showing relative differences – here with respect to the known 'truth' – would be insightful in the tropospheric region, i.e. to see the Complete Data Fusion performance where absolute values are small. Next to the supplementary

AMTD
material, why not add the average in Figure 2 as well?

- The histogram in Figure 4 seems to go up to 160 only, while Table 2 mentions 165 as a maximum in its latest column. Or the values in the histogram above 160 too low to be observed, or is there a different reason for this discrepancy?

- Lines 218 and following: Aires et al. (2012) have introduced a 'synergy factor' (not 'synergic factor' as stated in this work) for the errors only, while the authors have extended this concept to other quantities as well, including what they misleadingly call vertical resolution in Eq. (10) (see general comments). The authors should better explain the rationale behind the extension of the synergy factor to other quantities.

- Lines 261-262: "This is likely caused by the coincidence errors that have to be added in the fusion process" is a too brief and unsatisfactory explanation. With full control over the simulation and its errors, this should be quantitatively examined. Moreover, only the middle to upper stratosphere is mentioned, while Figure 6 clearly has values below one in the lower troposphere as well. - Line 294: Level 3 products are often provided on monthly timescales. It would be insightful to include a note on the use and (representativeness) effects of Complete Data Fusion in a large temporal domain.

- The authors seem to have somewhat exaggerated in their self-referencing: Ceccherini et al., 2003 and Ceccherini et al., 2010 do not seem to be required upon using Rodgers, 2000 already. Moreover, Ceccherini et al., 2014 is listed in the references, but not in the text. Finally, Kroon et al., 2011 is not required after Liu et al., 2010 (line 84).

Technical corrections:

- A single paragraph abstract would improve readability. Also throughout the text and conclusions, very often very short paragraphs are used. Several of these could be combined for clarity.

- Line 23: "and is therefore justifiable only as a temporary solution" is a user decision and irrelevant for this work.
- Line 24: It looks as if "while" is missing between "dataset" and "limiting"?

- Lines 29-30: What errors (or uncertainties) are referred to here?

- Introduction, first sentence: The Copernicus program contains more than only the Sentinel missions, so the provided web link should be at the end of the sentence, possibly by the introduction of a second link for the Copernicus program. The program moreover is an initiative of the European Commission, not of the European Union.

- Line 64: Remove "with each other".

- Using sub-numbering (a, b, c...) in Eq. (6) would be helpful. Providing alpha\_i and S-tilde\_i (two last equations) before the four others with some additional clarification could help very much in understanding the Complete Data Fusion setup.

- Figure 2 and comparisons in supplementary material: What the authors call "soliddot" and "dashed" lines actually both refer to "dash-dotted" lines as they are typically called.

- Line 192: What is the latitude-longitude range of the large domain? Repeating the single grid domain in the section title is misleading here.

- The averaging kernel matrix in the denominator of Eq. (9) should have index i,ll instead of f,ll.

- Line 297: The availability of the climatology and MERRA2 data should be mentioned as well.

- Equations, figures, tables, and (lack of) section numbering do not (yet) follow AMT(D) guidelines.

AMTD

---

## Author Comment (AC1) · 16 Mar 2020

Florence, 16/03/2020

We thank the reviewer for the very useful comments. In the following, we answer the specific comments (included in "boldface" for clarity) and, whenever required, we describe the related changes that will be implemented in the revised manuscript. Page and line numbers indicated refer to the original version of the paper published on AMTD.

**COMMENT #1: First, in the "Results and discussion", starting from line 186 and into the conclusions, the authors assume that a DFS increase automatically implies an increased vertical resolution of the retrieval. This is not true, as a DFS change can also be induced by a change of the weight of the measurement (or, equivalently, of the prior information) in the retrieval, as represented by the averaging kernel matrix' row sums, or by a change in the retrieval height offset (vertical shift of the measurement weight). Both are independent of the retrieval's vertical resolution, which e.g. can be determined from a kernel's FWHM. It is very important that the authors consider these alternatives based on a thorough discussion of (fused) averaging kernel matrix behaviour and either modify their discussion and conclusions accordingly or demonstrate that only the vertical resolution is impacted by the method.**

To answer this comment, we take as reference the case presented in the paragraph entitled "**Single grid-box analysis (0,5°x0,625°)**" rows 164-190, Figures 1, 2 and 3. In the right panel of Fig. R1 (new version of Fig. 3 in the original manuscript) the vertical resolution profile of the FUS product is compared with the vertical resolution profiles of the 118 L2 fusing products in the considered grid box cell, where the vertical resolution is calculated starting from AK matrices according the Full Width Half Height approach (Rodgers, 2000) and, in particular, using the algorithm defined in (Ridolfi and Sgheri, 2009). According to this algorithm the vertical resolution at a given vertical level is calculated as the ratio of the area under curve defined by the module of the AKM row whose diagonal value (i.e. the value that lies on the AKM diagonal) correspond to the considered level and the diagonal value of the same row; according to this particular formulation the presence of secondary lobes of the AKM row degrades the vertical resolution. In the figure, we can see that, even if the vertical resolution of the FUS product is almost everywhere improved with respect to the L2 fusing products, this does not happen in the range 30-45 km.

[Figure]

**Fig. R1 (Fig3 new):** *Left panel: AKs diagonal of: S4-TIR products (red lines), S5-TIR products (blue lines) S4-UV products (red lines) and FUS product (dark red line). In the text box, the average number of DOFs for each type of L2 product, the average number of DOFs for all L2 products and the number of DOFs of the FUS product are reported. **Right panel**: Vertical resolution (FWHH) profiles of: S4-TIR products (red lines), S5-TIR products (blue lines) S4-UV products (red lines) and FUS product (dark red line).*

To better understand the effect of the fusion on the AK matrices is useful to analyse the behaviour of their individual rows. In Fig. R2 two rows are represented, one that refers to the troposphere (left panel, 6 km) one to the middle stratosphere (right panel, 39 km), where the indicated reference altitude is the one corresponding to the value of the row that lies on the diagonal

of the AK matrix. The value of the vertical resolution at the considered altitude is reported in the legend while the diagonal value of each row is evidenced in the graphs with cross (L2) or dot (FUS) markers. At lower altitudes (left panel), as suggested by the reviewer, the DOFs increase can be attributed to three distinct phenomena where the first is the shrinkage of the main FUS AK lobe and the consequent improvement (of more than 30%) of the vertical resolution with respect to L2 products. The second phenomenon is linked to the fact that while for the FUS product the maximum value of the AK row corresponds to its diagonal element, for the L2 products these maxima are shifted with respect to the reference altitudes. The last phenomenon is a stronger contribution of the measurements with respect to the a priori in the FUS product, where the latter effect can be evidenced considering the sum of all the elements of the rows that has 0.913 as maximum value for the L2 products and is 0.956 for the FUS product. In this particular case, all these three effects go in a direction that can be regarded as a benefit of CDF application. The results at higher altitudes (39 km, right panel) are primarily influenced by the shape of the AK rows that exhibit large secondary lobes degrading the vertical resolution. Both figures and their analysis will be reported in the reviewed paper.

[Figure]

**Fig. R2 (Fig.4 new):** *Rows of AK matrices at 6 km (**Left panel**) and at 39 km (**Right panel**) for: S4-TIR products (red lines), S5-TIR products (blue lines) S4-UV products (red lines) and FUS product (dark red line). The value of the vertical resolution at the reference altitude is reported in the legend (the minimum of vertical resolution at the considered vertical level for each type of L2 product) while the diagonal value of each row is evidenced in the graphs with cross (L2) and dot markers (FUS).*

**COMMENT #2 The averaging of profiles as a simple combination technique is not mentioned in the abstract nor the introduction, while it is being discussed in a separate section eventually. Mentioning a discussion of the Complete Data Fusion method with respect to simple averaging from the beginning would improve readability.**
Mentions will be added in abstract and introduction.

**COMMENT #3 Stating "the quality of the products improving with larger grid boxes" in the abstract is misleading, as representativeness errors are also expected to increase with the grid box size. The latter is only briefly mentioned in the very last sentence before the conclusions. This important point deserves more discussion (especially regarding the differences in natural variability for different atmospheric molecules) and mentioning in the abstract.**
The sentence "the quality of the products improving with larger grid boxes" will be deleted from the paper. The point of representativeness errors will be more mentioned and discussed even if a dedicated study is needed to analyse the problem.

**COMMENT #4 Line 45: "whenever the user does not need the full spatial and temporal resolution" sounds misleading, as it seems that the Complete Data Fusion method can also be used to combine measurements from several satellites choosing e.g. the pixel size of one of the contributing instruments?**

In the considered examples the coincidence grid cell is much larger than the footprint of the L2 fusing products that have homogeneous pixel sizes. On the other hand, when the CDF is applied to L2 products with very different horizontal footprint size, the largest pixel footprint can be chosen as coincidence grid cell, so we do not think that the cited sentence is misleading. Nevertheless, if the reviewer believes this may lead to misunderstandings, we will consider the option to modify or to remove the sentence.

**COMMENT #5 Lines 75-77: The specific differences and usages of the ozone climatology and the atmospheric scenario might be far from clear for some readers. It would be very helpful to extend on this and relate this information with the quantities in the equations.**

References to the equations added.

**COMMENT #6 The sentence (paragraph) on line 120 is very unclear (or trivial) and fully without context.**

Sentence removed.

**COMMENT #7 Lines 132-133: "the diagonal elements of S_coinc are calculated as the 5% of profile estimates in the ozone climatology" is not clear at all.**

"When CDF is applied to not perfectly coincident products, the diagonal elements of $\mathbf{S}_{coinc}$ are calculated as the square of the 5% of the a priori profile $\mathbf{x}_a$."

**COMMENT #8 Lines 135-137: It is agreed that interpolation errors do not apply in this study, but as a demonstration "for a Full Exploitation of Copernicus Atmospheric Sentinel Level 2 Products" (title) the implications of the need for a preceding interpolation for upcoming reality should nevertheless be decently discussed.**

The following discussion will be added to the reviewed paper. "The formulas of Eqs.(6) refer to the case of measurements made on the same vertical grid. In general, also an interpolation error may be needed considering that the retrievals of the products to be fused can be furnished on different vertical grids. In (Ceccherini et al. 2018) the general expressions of CDF in the case of the fusion of products characterized by different vertical grids is presented and discussed together with the expression of the interpolation error that depends on the involved grids and on the AK matrices of the fusing products. However, since the interpolation error does not apply to the present study (the L2 products have been simulated on the same vertical grid) it has not been considered in Eqs. (6) and in the following discussion."

**COMMENT #9 - Line 153: The authors could refer to van Clarmann and Glatthor, 2019, for possibly improving their discussion on the averaging kernel matrix of an average product.**

The reference will be added.

**COMMENT #10 - Lines 158-161: This paragraph is way too short for a proper understanding of how the simulated retrievals are spatiotemporally allocated. How are instrument and orbit characteristics taken into account and applied to a past atmospheric scenario?**

The L2 datasets have been generated according to the equations (1)-(5) described in paragraph 2.2. The details of the simulation process (how are instrument and orbit characteristics taken into account) can be explored in the technical note (Technical Note On L2 Data Simulations, 35 pp., https://www.spacehatch.eu/result/616192, 2017) considering that here we simulated all the

pixels corresponding to a clear sky line of sight in the atmospheric scenario without applying any additional selection criteria. In fact, in the AURORA project 4 months of data have been considered, but a subset of the clear sky pixels has been simulated to reduce the computational cost of the simulations. For this study, all the clear sky pixels in the considered hour of data have been simulated, without additional filtering, choosing the reference time interval so that S4-S5 coincidences occur; the spatial distribution of the simulated products is indirectly represented in the left panel of Fig.5 that reports the horizontal distribution of FUS types (see Tab. 2). This paragraph will be added in the reviewed paper.

**COMMENT #11 - Figure 2 and comparisons in supplementary material: Although absolute differences have their benefits, (additionally) showing relative differences–here with respect to the known 'truth' – would be insightful in the tropospheric region, i.e. to see the Complete Data Fusion performance where absolute values are small. Next to the supplementary material, why not add the average in Figure 2 as well?**

A supplementary panel will be added to Fig.2, showing relative differences (see Fig.R3).

[Figure]

**Fig.R3 (Fig2 new):** *(Left panel): absolute differences between L2 profiles and their true profiles (green lines), absolute difference between the fused profile and the average of the true profiles (dark red continuous line), the average of $\sigma_{total}$ of L2 simulations (black dash-dotted lines), $\sigma_{f\,total}$ (dark red dash-dotted lines). (Right panel): relative percentage differences between L2 profiles and their true profiles (green lines), relative percentage difference between the fused profile and the average of the true profiles (dark red continuous line), the average of $\sigma_{total}$ of L2 simulations normalized with respect to the true profile and expressed in percentage (black dash-dotted lines), $\sigma_{f\,total}$ normalized with respect to the true profile and expressed in percentage (dark red dash-dotted lines).*

We prefer not to add the average in Fig.2 because too many lines can confuse the reader. On the other hand, we have motivated why we do not consider the average in the paragraph "Arithmetical average and biases" of the paper and in the paragraph entitled "Fusion of 1000 pixels in coincidence" in the supplementary material with particular reference to the right panel of **Fig. S1**.

**COMMENT #12 The histogram in Figure 4 seems to go up to 160 only, while Table 2 mentions 165 as a maximum in its latest column. Or the values in the histogram above 160 too low to be observed, or is there a different reason for this discrepancy?**

There was an error in the code generating the histogram so that the bars stopped at 160 even if there were data between 160 and 165. Graph corrected.

**COMMENT #13 Lines 218 and following: Aires et al. (2012) have introduced a 'synergy factor' (not 'synergic factor' as stated in this work) for the errors only, while the authors have extended this concept to other quantities as well, including what they misleadingly call vertical resolution in Eq. (10) (see general comments). The authors should better explain the rationale behind the extension of the synergy factor to other quantities.**

Corrected "synergic" with "synergy".

After the reference to Aires et al., the following paragraph has been added: "Although Aires introduces SF only for errors (Eq. (11)), we extended here his definition also for other quantities because they constitute a useful tool to synthetically represent the performances of fusion algorithms.

We modified the comment to Eq.10 as follows: "A value of *SF AK* larger than 1.0 at a specific vertical level (indicated by the index *l*) means that, at that level, the diagonal value of the AK matrix of the FUS product is higher than that of all the individual products. As we have seen commenting Fig.3 (new) and Fig.4 (new), the increase of the AK diagonal values at a specific level can happen for different reasons but all of them can be considered as an improvement in the product quality."

**COMMENT #14 Lines 261-262: "This is likely caused by the coincidence errors that have to be added in the fusion process" is a too brief and unsatisfactory explanation. With full control over the simulation and its errors, this should be quantitatively examined. Moreover, only the middle to upper stratosphere is mentioned, while Figure 6 clearly has values below one in the lower troposphere as well.**

This aspect has been studied in more detail and explained as follows.

Some SF AK values, both in the troposphere and in middle upper stratosphere, are smaller than one: in the troposphere, this happens in 20 cells out of 1939 while in the middle upper stratosphere this happens in almost 500 cells. In both cases, this is caused by two main simultaneous reasons: the first one (and the easier to explain) is the introduction of the coincidence VCM, which degrades the quality of the AKM. This effect is represented in Fig. R4 where the profiles of SF AK are represented for a single cell in which 56 L2 products have been fused, considering the fusion both with the coincidence error added and without. It can be noted that the introduction of the coincidence error provokes a SF AK values smaller than one in the troposphere.

[Figure]

**Fig. R4**: *profiles of SF AK for a single cell in which 56 L2 products have been fused, considering the fusion both with the coincidence error added (dark red line) and without (orange dashed line).*

The second reason for these less than one SF AK values derives from the application of the CDF to cases in which one of the fusing products has the peak of the AKM row that is much larger and closer to the nominal vertical level than all the others. In Fig. R5, we represent the rows of the AK matrices referring to an altitude of 3 km for a cell in which two L2 products in perfect coincidence (S4-TIR and S4-UV) have been fused.

[Figure]

**Fig. R5:** *rows of the AK matrices referring to an altitude of 3 km for a cell in which two L2 products have been fused; FUS product (dark red line), S4-TIR product (red dashed line), S4-UV product (green dashed line).*

It can be noted that the peaks of the rows of the fusing products happen at 12 km (S4-UV, red line) and 6 km (S4-TIR, green line) while the peak of the FUS product row is located at 9 km. Even if the peak of the FUS row is the largest one, since it is more distant from 3 km with respect to the S4-TIR peak, the diagonal value of the row of the FUS product (dark red dot) is smaller than the one of the S4-TIR product (green cross) so that the SF AK is less than 1 at 3km. A summary of this discussion will be reported in the reviewed paper.

**COMMENT #15 - Line 294: Level 3 products are often provided on monthly timescales. It would be insightful to include a note on the use and (representativeness) effects of Complete Data Fusion in a large temporal domain.**
We have not yet investigated thoroughly the application of CDF to long time averages and we can deepen this aspect only to a limited extent. The application of CDF requires the introduction of a coincidence VC matrix that is needed to manage the variability of the fusing products and that consequently depends from the size in time and space of the considered coincidence grid cell. The fused product can be considered as an estimate of the average of the "true" quantity in the coincidence grid cell. When a monthly timescale or an entire latitude band are considered, since a big number of L2 products are fused together, the contribution of the a priori will be less and less important and the choice of the "right" amount (not too much and not too little) and the "shape" of the coincident error have to be studied in depth in a dedicated study. A sentence about this issue will be added in the conclusions of the reviewed paper.

**COMMENT #16 - A single paragraph abstract would improve readability. Also, throughout the text and conclusions, very often very short paragraphs are used. Several of these could be combined for clarity.**
In the reviewed paper, the abstract will be formatted as a single paragraph and the paragraphs combined throughout the text.

**COMMENT #17 The authors seem to have somewhat exaggerated in their self-referencing: Ceccherini et al., 2003 and Ceccherini et al., 2010 do not seem to be required upon using Rodgers, 2000 already. Moreover, Ceccherini et al., 2014 is listed in the references, but not in the text. Finally, Kroon et al., 2011 is not required after Liu et al., 2010 (line 84).**

Ceccherini et al., 2003, Ceccherini et al., 2010, Ceccherini et al., 2014 and Kroon et al., 2011 removed.

**COMMENT #18 - Line 23: "and is therefore justifiable only as a temporary solution" is a user decision and irrelevant for this work.**

Eliminated.

**COMMENT #19 - - Line 24: It looks as if "while" is missing between "dataset" and "limiting"?**

Corrected.

**COMMENT #20 - - Lines 29-30: What errors (or uncertainties) are referred to here?**

Total errors, one of the sentences will be removed.

**COMMENT #21 - -Introduction, first sentence: The Copernicus program contains more than only the Sentinel missions, so the provided web link should be at the end of the sentence, possibly by the introduction of a second link for the Copernicus program.**

Corrected.

**COMMENT #22 - The program moreover is an initiative of the European Commission, not of the European Union.**

Corrected.

**COMMENT #23 -- Line 64: Remove "with each other".**

Removed

**COMMENT #24 -- Using sub-numbering (a, b, c...) in Eq. (6) would be helpful. Providing alpha_i and S-tilde_i (two last equations) before the four others with some additional clarification could help very much in understanding the Complete Data Fusion setup.**

Equations reordered. Sub numbering added.

**COMMENT #25 -- Figure 2 and comparisons in supplementary material: What the authors call "soliddot" and "dashed" lines actually both refer to "dash-dotted" lines as they are typically called.**

Corrected.

**COMMENT #26 -Line 192: What is the latitude-longitude range of the large domain?**

The spatial distribution of the simulated products is represented in the left panel of Fig. 5 that reports the horizontal distribution of FUS types (see Tab.2).

**COMMENT #27 - Repeating the single grid domain in the section title is misleading here.**

Removed.

**COMMENT #28 - - The averaging kernel matrix in the denominator of Eq. (9) should have index i,ll instead of f,ll.**

Corrected.

**COMMENT #29 - Line 297: The availability of the climatology and MERRA2 data should be mentioned as well.**

Data availability added.

**COMMENT #30 - Equations, figures, tables, and (lack of) section numbering do not (yet) follow AMT(D) guidelines.**

Section numbering added. Figures and tables moved at the end of the paper. Captions formatted according to AMT(D) styles.

---

## Referee Comment (RC2) · Anonymous Referee #2 · 4 May 2020

The paper by Zoppetti et al. presents the application of a so-called "Complete Data Fusion" approach to derive a single ozone vertical profile out of hundreds of satellite-derived ozone vertical profiles at pixel resolution to synthetic measurements of the Sentinel 4 and Sentinel 5 future missions. This is presented as a better alternative to a simple arithmetic average of the pixel-resolution derived ozone profiles, since vertical resolution or sensitivity of the retrieval is increased as compared to individual retrievals and also a reduction of the retrieval error is obtained. This concept is promising and the application of such an approach for future ozone retrievals is very interesting. However, the current manuscript dramatically lacks explanations of the results presented in the paper and also comparison to other approaches. Moreover, some important inconsistencies in the dataset used for the analysis is found, such the performance of one of the measurements (the infrared sensor of the Meteosat Third General or as mentioned in the paper S4 TIR sensor). The title of the paper is not appropriate either.

Major remarks that I strongly recommend to address:

1) Title of the paper: the current title is far too general, and it should be focused on the actual work presented on the paper. I strongly recommend clearly indicating the only species analyzed "atmospheric ozone profile", the only measurements used "Meteosat Third General and EUMETSAT Polar System Second Generation" and that is a sensitivity study based on simulations. I disagree indicating "Full Exploitation" since it is not the only way to exploit these measurements and "Copernicus Atmospheric Sentinel Level 2 Products" since only one species and 2 missions are concerned. By the way, Sentinel 4 and Sentinel 5 are the denomination of the UVNS sensors and not the ones operating in the infrared. The correct denotation is IRS onboard MTG-S and IASI-NG onboard EPS-SG. The whole manuscript should be corrected in that sense.

2) Introduction and scientific context of this work: this is one the major missing aspects in the paper. The instruction and the other section very rarely cite nor mention other related works on the subject different from previous works of the authors themselves. Although they do exist relatively abundantly, the authors do not mention any other approaches (except for the plain average) to use synergistically different measurements of the same or different sensors to derive ozone profiles. The authors should thoroughly provide a full exhaustive list of approaches of synergism of several measurements to derive ozone profiles and compare them (at least conceptually) to the proposed fusion approach.

3) Explanation of the results of the approach: only the equations of the fusion approach are transcribed in the paper, without any physical explanations, and the obtained results are very superficially described, with very little explanation for understanding them. Although this is the heart of the paper, the reader cannot understand

why the fusion approach allows an enhancement of vertical resolution or sensitivity to ozone from the synergism of several hundreds of profiles. This aspect should be thoroughly explained in physical terms and illustrations should be given. Concerning the retrieval errors, many explanations are missing, and this should be at least compared to the error of the average within the main text, in each occasion. Another missing aspect is how the authors represent systematic errors and co-locations errors. This is important to know and be explained in this current paper, since the persistance of such errors partly differentiates the current approach from the arithmetic average.

4) Datasets used in the paper: the explanation of the missions and simulated datasets is extremely brief. The reader does not understand what the differences between sensors are and why they provide certain DOFs or spatial coverage. The performance of the "S4:TIR" sensor does not seem to correspond to its instrumental characteristics. This sensor, which in reality is called IRS onboard MTG-S and not S4:TIR, has similar instrumental characteristics as IASI, with even a coarser spectral resolution and similar radiometric noise in the ozone band around 10 microns. The total column DOF for an ozone retrieval from IASI is typically 3 and at most 3.5. The current paper shows DOFs for "S4:TIR" greater than 5, which is not possible in my understanding. Unless thoroughly explained, justified and compared to IASI, all results considering "S4:TIR" simulated L2 products should be done again with proper instrumental characteristics. Moreover, the description of the atmospheric scenario should provide within this paper in much more detail (e.g. the resolution and particularities of pseudo-reality, sources, variability, etc).

More particular aspects:

5) Abstract: only tens or hundreds of measurements fall within tens of kilometers if satellite observations are finely resolved. This was not the case before Sentinel 5P for ozone retrievals and it is not the case for IASI-NG either.

6) L23: One can also average or do the median of the datasets.

7) L45-51: This is not true. By reducing the horizontal resolution, we lose natural variability within the grid cell.

8) Sections should be number in order to cite them.

9) L105-108: this sentence is not very clear. Please reformulate it clearly defining S and Stotal

10) L114: "The above formulation was used to simulate ozone profiles in the two spectral bands (UV and TIR) for both S4 and S5" please reformulate. Ozone profile are retrieved using measurements from a spectral band.

11) L120: However, these are not true retrieval from an iterative numerical procedure. How this formulation compares to true retrievals as those from true measurements? This aspect should be clarified and illustrated. Are S and Stotal consistent with those from true full retrievals?

12) L132: why 5%? This should be justified.

13) L139: the notion of "good" or bad is subjective. This cannot be expressed in such a why, but in objective terms (reduction of errors, bias, sensitivity, etc).. Please, reformulate.

14) L173: when embedded in the text, please use the word Figure and not Fig.

15) Error of fuse profile: it is not as low as 1 over the square root of the number of measurements as it would be for random errors and the arithmetic average but only around -30%. Comment thoroughly and explain.

16) Fig.3: It should be clearly written in the caption of the figure and the text that the AK of the "fuse profile" comes from the fusion of 118 profiles and the other AK are for single measurements.

17) Fig. 3: A full description of the 4 instruments and their characteristics should be given.

18) Fig. 3: S4 TIR: DOF of 5 is too high for the IRS sensor.

19) Fig. 3: Why only 4 curves are displayed instead of 5 AKs (4 sensors + fusion result)?

20) Fig. 3: Results not explained: DOF of 9.5 how do you explain this in physical terms? Where does it come from? Having only a few S5-derived profiles and more than a hundred of S4-derived, what Is the influence of having an asymmetric number of profiles from one or the other instruments?

21) Large domain section: the title indicates $0.5° \times 0.625°$ which is not large.

22) L193-L195: The sentence is not clear. Please reformulate. What is fusion grid-boxes?

23) L200: The explanation of variables should also provided in the caption of table 2

24) Please use Table and not Tab. Same Figure and not Fig. in the caption.

25) L195-198: What happened to S5-TIR? Why it is not here?

26) Table 2: this nomenclature is not clear S4:TIR+UV1_S5:TIR+UV1. Please reformulate here and elsewhere. What is the meaning of ":" and "_" in a name. They should be avoided in the names and the true names for the TIR sensors should be used.

27) Figure 4: the fraction of clear sky measurements seems very reduced. Although they exist in reality, no measurements with a small cloud fraction are considered? This should be clearly stated.

28) Why S5UV1 are only available over Northern Africa? Why we do not have S5 pixels near Greenland? This should be explained thoroughly in this paper since is a major dataset of the paper and not referred to previous papers.

29) L217 and L230: Why do you justify again avoiding the use of averaging without any quantified and clear statement. This method of arithmetic average should be explicitly

included in the comparison every time and compared in terms of error and performance.

30) L220: What is the meaning of pure number? Without units?

31) Fig.5 shows that SF DOF is at most 1.9 and this seems to be the case for Fig. 3. This means that the example of Fig.3 is not a typical case with SF DOF around 1.5 but the maximum performance of the fusion. The choice of this example should be reviewed, and typical case should be taken, but not the best.

32) I do not understand why the best performance is found for the use of the two S4 products, since the performance of the TIR sensor of S4 is not the best, as compared to that on S5.

33) Fig.5 caption: it should be explained that is the product of combining a large number of measurements. I recommend not to use N but Number in the axis label

34) Figure 6: What is the link between SF for AK and for Error? It seems that large SF AK correspond to smaller SF Error and vice versa (looking at green and purple dots). This should be explained and clearly quantified.

35) Table 3: it seems strange that no results of the $1 \times 1°$ cells are provided in the text. This should be commented and at least provided in terms of a table and compared to the smaller cells.

36) L293-295: I do not think that this is not true. The fusing products come from L2 products; they are intrinsically dependent in a priori information of these products.

---

## Author Comment (AC2) · 26 Jun 2020

Florence, 24/06/2020

We thank the reviewer for the very useful comments. In the following, we answer the specific comments (included in "boldface" for clarity) and, whenever required, we describe the related changes that will be implemented in the revised manuscript. Page and line numbers indicated refer to the original version of the paper published on AMTD.

**1) Title of the paper: the current title is far too general, and it should be focused on the actual work presented on the paper. I strongly recommend clearly indicating the only species analyzed "atmospheric ozone profile", the only measurements used "Meteosat Third General and EUMETSAT Polar System Second Generation" and that is a sensitivity study based on simulations. I disagree indicating "Full Exploitation" since it is not the only way to exploit these measurements and "Copernicus Atmospheric Sentinel Level 2 Products" since only one species and 2 missions are concerned. By the way, Sentinel 4 and Sentinel 5 are the denomination of the UVNS sensors and not the ones operating in the infrared. The correct denotation is IRS onboard MTG-S and IASI-NG onboard EPS-SG. The whole manuscript should be corrected in that sense.**

We agree that the title is too general so we propose a new one that is focused on the actual work: "*Application of the Complete Data Fusion to the ozone profiles measured by the Copernicus Atmospheric Sentinel missions: a feasibility study*" This new title aims also to address another possible source of misunderstanding that probably originates some of the referee comments. This study is focused on the CDF performances and the particular case and species addressed can be considered examples. In other words, the primary objective of this study is to show that, if the considered L2 products are profiles retrieved with optimal estimation techniques and if all the needed quantities are available, then the CDF is, in general, a valuable alternative.

The fact that the L2 products have been simulated in realistic conditions is an important supplementary benefit of the article that allows to quantitatively estimate the benefits of the CDF application, but in any case, is of secondary importance with respect to the feasibility study in general.

**2) Introduction and scientific context of this work: this is one the major missing aspects in the paper. The instruction and the other section very rarely cite nor mention other related works on the subject different from previous works of the authors themselves. Although they do exist relatively abundantly, the authors do not mention any other approaches (except for the plain average) to use synergistically different measurements of the same or different sensors to derive ozone profiles. The authors should thoroughly provide a full exhaustive list of approaches of synergism of several measurements to derive ozone profiles and compare them (at least conceptually) to the proposed fusion approach.**

Two new paragraphs have been added to the introduction mentioning and conceptually comparing different approaches for the synergistic use of the same or different sensors to derive ozone profiles and 5 new related references have been added to the bibliography.

**3a) Explanation of the results of the approach: only the equations of the fusion approach are transcribed in the paper, without any physical explanations, and the obtained results are very superficially described, with very little explanation for understanding them. Although this is the heart of the paper, the reader cannot understand why the fusion approach allows an enhancement of vertical resolution or sensitivity to ozone from the synergism of several hundreds of profiles. This aspect should be thoroughly explained in physical terms and illustrations should be given.**

Even if the presentation of the CDF is not the focus of the paper the fusion equations have been rearranged to be more readable and some comments have been added.

Regarding the presentation of the results, the section 3.2. "Single grid-box analysis (0.5°x0.625°)" has been extended and a new figure has been added to study in finer detail the enhancement of vertical resolution. More explanations have been also added to section 3.3. "Statistical analysis for a large domain".

**3b) Concerning the retrieval errors, many explanations are missing, and this should be at least compared to the error of the average within the main text, in each occasion. Another missing aspect is how the authors represent systematic errors and co-locations errors. This is important to know and be explained in this current paper, since the persistance of such errors partly differentiates the current approach from the arithmetic average.**

One entire section of the paper (2.4. "Arithmetical average and biases") and one section of the supplementary material ("Fusion of 1000 pixels in coincidence") have been dedicated to explain why the authors do not consider the error of the average in the remaining part of the paper. Regarding the systematic errors, the authors agree that this is an important point that has to be deeply studied in a dedicated work, possibly dealing with the application of CDF to real data. However, this paper is based on synthetic data and the systematic errors have not been considered. This is explained in the introduction (L56-61) where it is also mentioned how systematic errors can be treated in the CDF context. The interpolation error is not applied here but all the details concerning it and the coincidence errors can be found in two dedicated papers reported in the bibliography (Ceccherini et al. 2018 and 2019).

**4a) Datasets used in the paper: the explanation of the missions and simulated datasets is extremely brief. The reader does not understand what the differences between sensors are and why they provide certain DOFs or spatial coverage. The performance of the "S4:TIR" sensor does not seem to correspond to its instrumental characteristics. This sensor, which in reality is called IRS onboard MTG-S and not S4:TIR, has similar instrumental characteristics as IASI, with even a coarser spectral resolution and similar radiometric noise in the ozone band around 10 microns. The total column DOF for an ozone retrieval from IASI is typically 3 and at most 3.5. The current paper shows DOFs for "S4:TIR" greater than 5, which is not possible in my understanding. Unless thoroughly explained, justified and compared to IASI, all results considering "S4:TIR" simulated L2 products should be done again with proper instrumental characteristics.**

A new paragraph has been added at the end of section 2.2 L2 Product Simulation with a more detailed explanation of the missions and of the simulated datasets, with some more references and with explicit mention of relevant details.

Regarding the DOFs of TIR:S4 (i.e., IRS/MTG-S), the paper by (Crevoiser et al., 2014) shows that the number of DOSF is 4.7, when the IASI-NG instrument configuration is IRS2b, which was used for this study. Moreover, for generation of the synthetic retrieval products we used climatological a priori information rather than state of the art ozone information with the result of applying a weaker constraint that further increases the DOFs.

Ref:: Crevoisier, C., Clerbaux, C., Guidard, V., Phulpin, T., Armante, R., Barret, B., Camy-Peyret, C., Chaboureau, J.-P., Coheur, P.-F., Crépeau, L., Dufour, G., Labonnote, L., Lavanant, L., Hadji-Lazaro, J., Herbin, H., Jacquinet-Husson, N., Payan, S., Péquignot, E., Pierangelo, C., Sellitto, P., and Stubenrauch, C.: Towards IASI-New Generation (IASI-NG): impact of improved spectral resolution and radiometric noise on the retrieval of thermodynamic, chemistry and climate variables, Atmos. Meas. Tech., 7, 4367–4385, https://doi.org/10.5194/amt-7-4367-2014, 2014.

**4b) Moreover, the description of the atmospheric scenario should provide within this paper in much more detail (e.g. the resolution and particularities of pseudo-reality, sources, variability, etc).**

The section 2.1. "Atmospheric scenario and ozone climatology" has been extended and in particular the quantities taken from these external databases have been put in relation with the other quantities defined in the equations that describe the simulation process. The details on the atmospheric scenario goes beyond the scope of this paper and can be found in the cited references.

**5) Abstract: only tens or hundreds of measurements fall within tens of kilometers if satellite observations are finely resolved. This was not the case before Sentinel 5P for ozone retrievals and it is not the case for IASI-NG either.**

"in the near future" added at the end of the sentence.

**6) L23: One can also average or do the median of the datasets.**

The mention of the use of simple averages has been added to the new abstract.

**7) L45-51: This is not true. By reducing the horizontal resolution, we lose natural variability within the grid cell.**

The premise "whenever the user does not need the full spatial and temporal resolution" was originally introduced to prevent this objection. The information content intended here is the one represented by the DOFs, but since at L45 they have not yet been defined in the paper we preferred this more general formulation. Nevertheless, since the sentence is not strictly necessary we removed it.

**8) Sections should be number in order to cite them.**

Section numbering added.

**9) L105-108: this sentence is not very clear. Please reformulate it clearly defining S and Stotal**

The definitions of S and Stotal have been separated in two distinct paragraphs.

**10) L114: "The above formulation was used to simulate ozone profiles in the two spectral bands (UV and TIR) for both**

**S4 and S5" please reformulate. Ozone profile are retrieved using measurements from a spectral band.**

In this study ozone profiles have not been retrieved using simulated spectral measurements. Added "In this study".

**11) L120: However, these are not true retrieval from an iterative numerical procedure. How this formulation compares to true retrievals as those from true measurements? This aspect should be clarified and illustrated. Are S and Stotal consistent with those from true full retrievals?**

This method uses a linear approximation (Eq. (1)) of the relationship between the retrieved profile and the true profile in the optimal estimation method (see Rodgers, 2000). Therefore, the validity of this method is within the correctness of this approximation. A sentence clarifying this aspect is added.

**12) L132: why 5%? This should be justified.**

The value of 5% is a reasonable value. However, the study presented in (Ceccherini et al. 2019) clearly show that even if the coincidence error is strictly needed for the correct behaviour of the CDF product, this is not strongly dependent on its exact amount. Therefore, this choice does not affect the conclusions of the study. A paragraph has been added in the paper.

**13) L139: the notion of "good" or bad is subjective. This cannot be expressed in such a why, but in objective terms (reduction of errors, bias, sensitivity, etc). Please, reformulate.**

Removed "good".

**14) L173: when embedded in the text, please use the word Figure and not Fig.**

OK.

**15) Error of fuse profile: it is not as low as 1 over the square root of the number of measurements as it would be for random errors and the arithmetic average but only around -30%. Comment thoroughly and explain.**

Explanation added in section 2.3 right after eqs 6. This aspect is also studied in the supplementary material commenting Figure S1, right panel.

**16) Fig.3: It should be clearly written in the caption of the figure and the text that the AK of the "fuse profile" comes from the fusion of 118 profiles and the other AK are for single measurements.**

Sentence added in the caption.

**17) Fig. 3: A full description of the 4 instruments and their characteristics should be given.**

We have expanded the instrument description, see also answer to point 4, above.

**19) Fig. 3: Why only 4 curves are displayed instead of 5 AKs (4 sensors + fusion result)?**

As explained in the beginning of the section, no S5-UV1 products falls in the considered cell so only 4 curves are displayed. This choice also aims to simplify as far as possible the understanding of the discussion and the figures. The discussion of a case with 4+1 curves has been added to the supplementary material in the section "Single grid-box analysis (1°x1°)".

**20a) Fig. 3: Results not explained: DOF of 9.5 how do you explain this in physical terms? Where does it come from?**

The CDF acts removing the a priori information of the L2 fusing measurements and adding an a priori information (independent of that of the input measurements) in the fusion process. This characteristic of the CDF allows to increase the relative weight of the information coming from the measurements with respect to the information coming from the a priori, as a consequence we obtain an increase of the number of DOFs. This mechanism does not occur in the arithmetic average.

The section 3.2. "Single grid-box analysis (0.5°x0.625°)" has been extended and a new figure has been added to study in finer detail the enhancement of vertical resolution observing the behaviour of the AK matrices rows.

**20b) Having only a few S5-derived profiles and more than a hundred of S4-derived, what is the influence of having an asymmetric number of profiles from one or the other instruments?**

This is really an interesting point that cannot be explained in few words. A section has been added to the supplementary material entitled "Single grid-box analysis (1°x1°)" that accounts for both this aspect and the effect of enlarging the coincidence cell size. This analysis has not been included in the paper since we believe that it is too detailed for a feasibility study based on simulated products. On the other hand, we agree with the reviewer that these aspects are crucial and worth a throughout investigation in the case of the application of the CDF to real measurements.

**21) Large domain section: the title indicates 0.5° x 0.625° which is not large.**

The indication of the cell size in the title was misleading so it has been removed.

**22) L193-L195: The sentence is not clear. Please reformulate. What is fusion grid-boxes?**

Sentence modified. See also previous point (the misleading indication of grid box size in the title could also contribute to the confusion).

**23) L200: The explanation of variables should also provided in the caption of table 2**

Caption modified.

**24) Please use Table and not Tab. Same Figure and not Fig. in the caption.**

OK

**25) L195-198: What happened to S5-TIR? Why it is not here?**

L195-198 do not list all the types of product; S5-TIR is simply not mentioned here because not relevant in the discussion.

**26) Table 2: this nomenclature is not clear S4:TIR+UV1_S5:TIR+UV1. Please reformulate here and elsewhere. What is the meaning of ":" and "_" in a name. They should be avoided in the names and the true names for the TIR sensors should be used.**

The nomenclature of table 2 is the same used in the figures. The use of the real name of the sensors is too long for the legends in the figures, therefore, we prefer to maintain that nomenclature that, although not precise, is coherently used throughout the text. On the other hand, that nomenclature is explained in the "Description" column of table 2 itself.

**27) Figure 4: the fraction of clear sky measurements seems very reduced. Although they exist in reality, no measurements with a small cloud fraction are considered? This should be clearly stated.**

The section 3.1. "Fusion in realistic spatial and temporal resolution conditions: the L2 Datasets" has been expanded to explain better these aspects.

**28) Why S5UV1 are only available over Northern Africa? Why we do not have S5 pixels near Greenland? This should be explained thoroughly in this paper since is a major dataset of the paper and not referred to previous papers.**

The section 3.1. "Fusion in realistic spatial and temporal resolution conditions: the L2 Datasets" has been expanded to explain better these aspects. In particular the products considered in this paper have been simulated considering the daylight, clear sky pixels belonging to one particular hour and one particular orbit that do not refer to clear-sky daylight pixels in Northern Africa and Greenland.

**29) L217 and L230: Why do you justify again avoiding the use of averaging without any quantified and clear statement. This method of arithmetic average should be explicitly included in the comparison every time and compared in terms of error and performance.**

The justification for not using the arithmetic average is explained both in formulas (section 2.4) both with a numerical example (supplementary material, section entitled "Fusion of 1000 pixels in coincidence").

**30) L220: What is the meaning of pure number? Without units?**

Yes, because it is a relative quantifier expressed by the ratio of two quantities with the same units.

**31) Fig.5 shows that SF DOF is at most 1.9 and this seems to be the case for Fig. 3. This means that the example of Fig.3 is not a typical case with SF DOF around 1.5 but the maximum performance of the fusion. The choice of this example should be reviewed, and typical case should be taken, but not the best.**

The SF DOF of the case represented in Figure3 is 1.77 and not the maximum value 1.9. Therefore, this case does not correspond to the best performance, and it was chosen because it represents well the phenomena that are discussed in the paper. The value of 1.77 has not to be compared with the global average of SF DOF but with the SF DOFs obtained in cells with a similar number of fused L2 products. The case of Figure 5 has been chosen also because it is relatively easy to be explained. A more complicated case has been added in the supplementary material section entitled "Single grid-box analysis (1°x1°)". On the other hand, figure 3 and 4 give an overall picture about fusion performances; a new discussion and a new figure in section 3.2 and the new final paragraph of section 3.3 give an insight on the less than one values of SF AK in the troposphere and in the middle upper atmosphere that is also the origin of the cluster of green points in Figure 5. We think that this new scenario represents a good compromise in which all the relevant aspects are addressed.

**32) I do not understand why the best performance is found for the use of the two S4 products, since the performance of the TIR sensor of S4 is not the best, as compared to that on S5.**

The performance here is evaluated in relative terms (all SFs are ratios) so the better performance of CDF does not necessarily take place when the quality of the L2 product is higher. On the contrary the best performances of the fusion in terms of Synergy Factors are obtained when many products with comparable quality are fused. See also the new section entitled "Single grid-box analysis (1°x1°)" in the supplementary material for a case dealing with L2 products with different quality.

**33) Fig.5 caption: it should be explained that is the product of combining a large number of measurements. I recommend not to use N but Number in the axis label.**

Figure 5 represent the SF DOFs for all the 1979 fused products. The term N has been used consistently throughout the text, figures and tables of the article to indicate the number of cells or products so we prefer to maintain the original nomenclature in this figure.

**34) Figure 6: What is the link between SF for AK and for Error? It seems that large SF AK correspond to smaller SF Error and vice versa (looking at green and purple dots). This should be explained and clearly quantified.**

We do not think that this phenomenon can be generalized: for example, this does not seem to happen at lower altitudes. As shown in the paper the behaviour of the AK matrices is not intuitive and deserves a detailed study to be explained (see point 31, above and new section 3.2 in the paper).

**35) Table 3: it seems strange that no results of the 1x1 ∘ cells are provided in the text. This should be commented and at least provided in terms of a table and compared to the smaller cells.**

Since the 1x1 case does not introduce significant new features with respect to the smaller cell, we decided to briefly cite it in the text and to document it in the supplementary material.

**36) L293-295: I do not think that this is not true. The fusing products come from L2 products; they are intrinsically dependent in a priori information of these products.**

The CDF removes (by means of AK matrices) the original a priori information from L2 products before to combine them so the fused product is effectively independent from these a priori. In other words, the a priori information of the fused product can be chosen independently from the ones of the L2 products.

---

## Author Response (AR2)

**Response to RC1**

The authors have done a great effort to answer all reviewer comments appropriately. This manuscript has significantly improved as a result. The writing style however can still be considered somewhat sloppy at some instances. Technical corrections are therefore suggested below, but the authors are encouraged to have the final version of their manuscript revised by a native English speaker.

**Lines 30-33: Rewrite for clarity. Statistical analysis of what? Start a new sentence afterwards: "The grid box size was also..."**

done

**Line 35: "coordinated by the European Commission"**

Done

**Line 48: "contributing missions" is unclear here. What is contributing to what? Other missions contributing to the Sentinels would be misleading...**

Removed

**Line 49: "encompass from" is a wrong wording.**

Replaced with include

**Line 53: "The three approaches differ…"**

done

**Line 60: Two times particular(ly).**

Removed particularly.

**Line 65: Mismatch between single and plural.**

corrected

**Line 65: "which collects all the available information content" sounds vague. Something like "which optimizes the DFS of the fused product" and including a reference to one of the technical papers?**

… in which the sensitivity increases and the error reduces (Ceccherini et al., 2015).

**Line 70: "the algorithm" – which one is intended here; or should this be plural, referring to all previously mentioned?**

Should be plural, even if in this article we consider only the first

**Line 77: "and discuss the use of the profiles average as fusion technique" - This addition is misleading, as if averaging is the main fusion technique. Suggestion: "and discuss the differences between CDF and mere averaging"**

done

**Lines 89-91: Duplication of following?**

89-91 represent a general description, in the following more detailed description is presented.

**Line 149: "specifications used for the simulation"**

"in" was removed

**Line 159: "Eqs."**

done

**Line 163: "Concerning the profile and the error the CDF…"**

done

**Lines 169-171: Could you briefly elaborate on how this 5% choice is made based on grid cell size? How should other users of the CDF method make such decisions for different grids? Could you provide a few references?**

The 5% choice matured in a heuristic way by varying its value and observing the quality of the fused product, both for some single cells chosen as a reference and by looking at the Synergy Factors of the entire dataset using similar representations to figures 6 and 7 (added in the text, lines 178-180).

The figure below (not in the paper) refers to the cell of figures 1-4, representing FUS total error profile, the L2 total error average profile, the 5% of the a priori profile and the standard deviation profile of the "virtual truth" in the cell of interest. Note here that the black error profile is about half of the green one and in any case always larger than the blue one at all heights. This is a confirmation of the following sentence in the paper (lines 184-186) : "… even if the coincidence error is strictly needed for the correct behaviour of the CDF product, this is not strongly dependant by its exact amount until it is smaller with respect to the errors of the individual L2 products". In the same paragraph we cite (Ceccherini et al. 2019) both to justify the heuristic approach (see above) and to suggest an alternative recipe to the $S_{coinc}$ choice.

[Figure]

**Eq. (8): Remove "= …"**

done

**Lines 208-209: "used in this article"**

done

**Line 231: "allows seeing"**

done

**Line 249: Remove "as suggested by one of the reviewers"**

removed

**Line 254: Replace "contribute of the measures" by something like "contribution of the (simulated) measurements"**

Replaced

**Response to RC2**

**The manuscript by Zoppetti et al. has improved in clarity for some of the aspects remarked by the reviewers. However, major revisions are still needed, and the manuscript cannot be published in the current state. My major concern is that there are mistakes in the characteristics of the instruments onboard MTG-S and EPS-SG together with the instruments Sentinel 4 and 5, whose measurements are simulated for demonstrating the performance of the Fusion algorithm. These instrumental characteristics are greatly important in this work. They must be corrected and all the results using these simulated measurements (Figures 3, 4, 6 and 7) should be done again with the correct configuration.**

Thanks to the reviewer's comments, we realized that there is a fundamental misunderstanding about the L2 simulated products characteristics and more in general on the paper objectives. Moreover, we realized that the specifications of the considered platform significantly changed from the one considered for simulations. In fact, we simulated L2 products in the context of an H2020 project that started in 2016 and ended in 2019 were the simulation was one of the first steps. The primary references for all the simulated products were the documents listed below, dated between 2007 and 2012.

- [D4] Jörg Langen, European Space Agency (ESA), (2007) GMES Sentinels 4 and 5 Mission Requirements Traceability Document (MRTD), EOP-SMA/1507/JL-dr, issue 1 rev.0. https://earth.esa.int/web/guest/document-library/browse-document-library/-/article/gmes-sentinels-4-and-5-mission-requirements-document-6442

- [D42] Mission Science Division, European Space Agency (ESA), (2012) GMES Sentinels 4 and 5 Mission Requirements Traceability Document (MRTD), EOP-SM/2413/BV-bv, issue 1 rev.0. http://aurora.ifac.cnr.it/utils/personaldocs/see/96/

- [D5] EUMETSAT, (2010) MTG End-User Requirements Document [EURD], EUM/MTG/SPE/07/0036, v3C https://www.ncdc.noaa.gov/sites/default/files/attachments/PDF_MTG_EURD.pdf

- [D6] European Space Agency (ESA), (2012) GMES Sentinels 4 and 5 Mission Requirements Document (MRD), EOP-SM/2413, issue 1 rev.0. http://aurora.ifac.cnr.it/utils/personaldocs/see/93/

- Crevoisier, C., Clerbaux, C., Guidard, V., Phulpin, T., Armante, R., Barret, B., Camy-Peyret, C., Chaboureau, J.-P., Coheur, P.-F., Crépeau, L., Dufour, G., Labonnote, L., Lavanant, L., Hadji-Lazaro, J., Herbin, H., Jacquinet-Husson, N., Payan, S., Péquignot, E., Pierangelo, C., Sellitto, P., and Stubenrauch, C.: Towards IASI-New Generation (IASI-NG): impact of improved spectral resolution and radiometric noise on the retrieval of thermodynamic, chemistry and climate variables, Atmos. Meas. Tech., 7, 4367–4385, https://doi.org/10.5194/amt-7-4367-2014, 2014.

The result is a set of L2 products that do not follow the updated specifications of the considered platforms. Since the project ended in 2019, it is not possible to repeat now the L2 product characterization and simulation process, as requested by the reviewer.

Nevertheless, this work focuses on a relative comparison of the fused and L2 products and on the ability of the Complete Data Fusion inducing quality improvements that are, in some sense, independent from precise instrumental characteristics.

On the other hand, we realize that these considerations have to be clearly stated in the introduction (paragraph added, lines 80-86) so that the reader can concentrate her/his attention on the more relevant aspect of this study.

In this sense, it is also opportune to change (again) the title of the article in "APPLICATION OF THE COMPLETE DATA FUSION TO THE OZONE PROFILES MEASURED BY GEOSTATIONARY AND LOW EARTH ORBITS SATELLITES: A FEASIBILITY STUDY".

In the following, we will answer to the other points raised by the reviewer using as reference the already cited documents and also the Requirements Document and the Technical Note cited below.

- [D2.1] AURORA consortium, (Advanced Ultraviolet Radiation and Ozone Retrieval For Applications, grant no. 687428): Requirements document Issue: 1, rev. 4 Date: 31-Mar-2016 http://aurora.ifac.cnr.it/utils/personaldocs/see/97/

- [D3.4] AURORA consortium, (Advanced Ultraviolet Radiation and Ozone Retrieval For Applications, grant no. 687428): Technical Note On L2 Data Simulations, 35 pp. http://aurora.ifac.cnr.it/utils/personaldocs/see/95/

**IRS onboard MTG-S (called in the paper S4-TIR): this sensor is NOT IASI-NG (nor IRS2b) which is described by Crevoisier et al. (2014). Crevoisier et al. (2014) do NOT describe the instrument that will be onboard the geostationary satellite MTG-S together with UVN/Sentinel 4, but they only describe IASI-NG which will be onboard the low orbit EPS-SG satellite together with Sentinel 5/UVNS. IASI-NG is supposed to be S5-TIR in the paper. EUMETSAT describes IRS onboard MTG-S here: https://www.eumetsat.int/website/home/Satellites/FutureSatellites/MeteosatThirdGeneration/MTGDesign/index.html "The Infrared Sounder (IRS) on MTG-S … with a hyperspectral resolution of 0.625 cm-1 wave-number..." This means that IRS onboard MTG-S (in your paper S4-TIR) will have an even coarser spectral resolution than IASI (0.5 cm-1) and much coarser resolution than IASI-NG (0.25 cm-1). The SNR for IRS onboard MTG-S will be much lower than that for IASI-NG and similar to IASI. IASI, with a finer spectral resolution than IRS/MTG-S, enables the retrieval of ozone with 3,5 DOFs at most. Therefore, it is not possible to obtain 5 degrees of freedom for the retrieval of the ozone total column derived from IRS onboard MTG-S, as illustrated in Figure 3 of the paper. In consequence, all simulations and retrievals using S4-TIR must be done again with the correct instrumental characteristics of this instrument. Figures 3, 4, 6 and 7 should be revised with correct simulations of IRS/MTG-S measurements.**

S4-TIR products are simulated according to [D4], [D42] and [D6] and their characteristics are summarized in [D3.4] from which we extracted the following table 8. In particular the NESR value of table 8 derived from (Crevoisier et al. 2014, IRS1b scenario). With these characteristics, the 5 degrees of freedom can be due to both the small error adopted and the relatively low a priori strength compared to operational retrieval conditions. As already said in the previous paragraph, it is not possible to repeat the simulations, and we also think that it is not worth to do. In this paper, we want to show the feasibility and the potential benefits of the a-posteriori fusion approach, and we think that the goal is reached even without an exact instrument characterization. Since we believe in achieving this objective, the next step of this research activity will be applying CDF to real data.

| | Description |
|---|---|
| *Instrument* | Infrared Sounder (IRS) |
| *O₃ retrieval spectral range (cm⁻¹)* | 1030-1080 |
| *Spectral resolution (cm⁻¹)* | 0.625 (apodized IRSF) |
| *Spectral sampling (cm⁻¹)* | 0.625 |
| *NESR* | IASI L1C VCM/4 (IASI apodized noise divided by 2) |

**Table 8:** S4 (GEO) TIR instrument characterization.

**IASI-NG (called in the paper S5-TIR): the spatial sampling for IASI-NG (called in your paper S5-TIR) is not correct. The spatial sampling for IASI-NG is the following (the same as for IASI): circular pixels of 12 km of diameters whose centers are spaced by 25 km at nadir. Figure 1 of the paper shows 5 pixels within 0.5 ° of latitude and that is not possible. One may have at best 3 pixels and typically 2 pixels within 0.5 ° (which is roughly 55 km) in both directions (along and across the satellite track). Therefore, it is not possible to have so many IASI-NG pixels within a box of 0.5° x 0.625°, but half of them. Therefore, the number of measurements used for the fusion for the given box is not correct. The spacing of the IASI-NG pixels must be corrected and the results should be revised.**

S5-TIR products are simulated according to [D4], [D42] and [D6] and their characteristics are summarized in [D3.4] from which we extracted the following table 4.

| | LEO (S5) | GEO (S4) |
|---|---|---|
| Field of view (km²) G/T | 5×5 / 12×12 | 5×5 / 15×15 |

**Table 4:** *Instrument characterization for LEO and GEO TIR measurements. Goal (G) and Threshold (T) values indicated in the table correspond, respectively, to estimates of the parameters in case the instrument performs at its best and to limit values that we expect to reach anyhow. AURORA will be using Threshold values for generation of simulated data.*

We assumed that the pixel had a square shape with a side of 12 km at nadir (and no spacing between them) and with that configuration, you can have about five pixels in latitude direction in a 0.5-degree grid box. The same general considerations of the previous two points apply here.

**UVNS (called in the paper S5-UV): The horizontal sampling of UVNS pixels is not correct either. As announced by ESA https://earth.esa.int/web/eoportal/satellite-missions/c-missions/copernicus-sentinel-5, the horizontal sampling of UVNS is 7 km x 7 km which is much smaller than the box of 0.5 x 0.625° in Fig. 1. Therefore, it is not possible that no pixel is available for UVNS (as indicated in the current version of the manuscript).**

S5-UV products are simulated according to [D4], [D42] and [D6] and their characteristics are summarized in [D2.1] from which we extracted the following table 3.2.4. Regarding the reviewer comment on 7x7 km pixel size: this refers to S5-UV2 (300-320 nm) band while S5-UV1 band (270 - 300nm) has pixel of about 50x50 km. Within AURORA, we took S5-UV1 band as reference target, and with a pixel size of 45x45 km it is possible not to have a pixel centre in a 0.5-degree grid box.

|  | LEO (S5) | GEO (S4) |
|---|---|---|
| Spectral range (nm) | 270-320 | 305-320 |
| spectral resolution (nm) | LEO-UV1: 1.0 (270-300nm) LEO-UV2: 0.5 (300-320nm) | 0.5 |
| Spectral sampling ratio | 3 | 3 |
| SNR radiance | LEO-UV1: 100 @ 270 nm LEO-UV2: 1000 @ 310 (G) / 320 (T) nm | 305 nm: 200 (G) / 160 (T) 310 nm: 400 (G) / 320 (T) 315 nm: 700 (G) / 630 (T) 320 nm: 900 |
| SNR irradiance | 10000 | 305 nm: 3000 (G) / 1000 (T) 310 nm: 3000 (G) / 1600 (T) 315 nm: 10000 (G) / 2000 (T) 320 nm: 10000 (G) / 2700 (T) |
| Spatial sampling distance | LEO-UV1: 50 (T) / 15 (G) km LEO-UV2: 15 (T) / 7 (B) / 5 (G) km | <= 8 km at 45°N and longitude of the satellite (0°E) |
| Geographical coverage | Continuous measurement on the illuminated side of the Earth's atmosphere, OZA <= 66°, performance requirements valid for SZA < 80°. | Longitude: 30°W – 40°E @ 40N Latitude: 25°N (G) / 30°N (T) – 65°N @ 0°E Rectangular in view angles, areas with OZA > 75° not required We ignore here any southward/northward shifts in winter/summer |

*Table 3.2.4: Instrument characterization for LEO and GEO UV measurements.*
*Goal (G) and Threshold (T) values indicated in the table correspond, respectively, to estimates of the parameters in case the instrument performs at its best and to limit values that we expect to reach anyhow. AURORA will be using Threshold values for generation of simulated data.*

**For avoiding all these errors, I strongly recommend using the real names of the instruments: UVNS/Sentinel 5, UVN/Sentinel 4, IASI-NG/EPS-SG and IRS/MTG-S, or at least part of their real names. In my opinion, creating new names that are only used in this paper only add confusion. In addition, "Sentinel" only refers to the UV-VIS instruments and not the TIR. Of course, the spectral resolution and horizontal sampling for each of them must be corrected.**

Since now it should be clear that we are referring to instruments whose characteristics are not aligned with the most recent specifications, we think it is appropriate to keep the terminology unchanged also to avoid generating further misunderstandings. To further help the reader, we added a new table (Table 1) that reports the more relevant instrumental parameters (cited in the previous answers) together with the adopted terminology.

**Another major point concerns the explanation of the results of Figure 3 of the manuscript. In the revision, a new paragraph explains the gain of sensitivity in the Fused product with respect to the L2 retrievals. However, they do not explain two key points: At the lowest point (1000 hPa) and near the highest (0.5 hPa) the AK for the fused product is much stronger than the ones of the L2 products. At 1000 hPa, a local minimum is seen for the AK of all the L2 products**

**(S4-TIR, S5-TIR and S4-UV), however the AK shows a relative maximum more than twice larger than the best case of the L2 products. At 0.5 hPa, the AK of the Fused product is 5 or 6 times larger than the individual L2 products. Elsewhere, the AK enhances by about 30 or 50 %, but not by a factor 2 or 6. Could you explain in the detail these features? Is this only a numerical feature of this single profile? What are the physical reasons for such an extremely large enhancement of sensitivity? and where would the information come from? Moreover, why vertical resolution in Fig. 3 right panel (FUS product) is the finest at 1000 hPa (above it is coarser)? This is never the case for any L2 product of ozone derived by optimal estimation or Tikhonov-Philips regularization. I cannot understand such a performance of the Fusion algorithm, which is not consistent with those of the original products. This should be clearly and thoroughly demonstrated.**

In the past revision, we added a new figure (Figure 4) to explain the enhancements obtained with the CDF in terms of vertical resolution, and even if this figure refers to different vertical levels (400 and 4 hPa) with respect to the ones that the referee mentions (1000 and 0.5 hPa) it represents the same effects noted by the reviewer.

In the paper, we recognized three distinct phenomena to which the DOFs increase can be attributed. The first is the constriction of the main FUS AK lobe and the consequent improvement of the vertical resolution with respect to L2 products. The second phenomenon relates to the fact that while for the FUS product the maximum value of the AK row corresponds to its diagonal element, for the L2 products these maxima are shifted with respect to the reference altitude of the rows. The last phenomenon is a stronger contribution of the measurements with respect to the a priori in the FUS product.

Here we repeat the same figure for the two vertical levels of interest. As in figure 4, at lower altitudes (1000 hPa, left panel) the three phenomena coexist: while for the FUS product the maximum value of the AK row corresponds to its diagonal element, the L2 maxima are shifted with respect to the reference altitude of the rows; also note that the peak of the FUS row tends to be narrower than the L2 ones resulting in a better vertical resolution. This is also the reason why the vertical resolution of the FUS product has its finest value at 1000 hPa, and this not happens to the L2 products. Considering the sum of all the value of the considered AK row we have 1.06 for the FUS and 0.82 for the L2.

[Figure]

At higher altitudes (0.3 hPa, right panel) the results also depend on the shape of the AK rows that exhibit large secondary lobes; here the main reason of the DOFs increase seems to be a stronger contribution of the measurements with respect of the a-priori. The large secondary lobes located at lower altitudes affect the values of the sum of the AK row (1.20 for FUS and vs 0.37 for L2).

These results depend mainly on the particular shape of the L2 AK rows so can be considered a numerical feature of this particular situation; on the other hand, the left panel of figure 7 (in the paper) and S7 (in the supplementary material) clearly show that they primarily depend on the FUS type and on the number of L2 fusing products. In these figures it is also evident that SF AK values larger than 10 are relatively common in the results relative to the considered dataset.

**Title: it is written "the Copernicus Atmospheric Sentinel missions" but it is not precise. The paper only refers to EPS-SG and MTG-S missions. There are other Sentinel missions that are used for the atmosphere which are not analyzed in the paper: Sentinel 5 Precusor/TROPOMI, Sentinel 3 SLSTR and there will probably be other atmospheric sentinel missions in the future. Therefore, the title should be revised: "the Copernicus Atmospheric Sentinel missions" should be replaced by EPS-SG and MTG-S missions.**

We propose the following new title: "APPLICATION OF THE COMPLETE DATA FUSION TO THE OZONE PROFILES MEASURED BY GEOSTATIONARY AND LOW EARTH ORBITS SATELLITES: A FEASIBILITY STUDY"

**Section 3.2: Please correct "contribute of the measures" … by "contribution of measurements"**

done

**Keep the same nomenclature for averaging kernels: AK or AKM, but do not use both.**

AKM removed

**Although requested in the previous review, authors provided only scarce additional information on "Atmospheric scenario" in section 2.1. A more detailed description of the used models should be provided.**

In the paper we provided the reference in which the atmospheric scenario is described and the reader, if interested, can deepen there the details of the used models. Moreover, we described the role of the scenario both in the L2 product simulation and in the CDF, also explicating the role of the virtual truth in the equations. On the other hand, we prefer not to report a more detailed description in the paper because not relevant to the main focus of the paper.

**This sentence and others are not correct "… ozone profiles measured in the UV region" Ozone is not measured in a spectral region. Please replace by "ozone profiles derived from measurements in the UV region"**

done.

**In the reply, please cite every change in the response to the reviewer's documents and the lines where they are. Otherwise, it is very difficult to track the changes done.**

Done (see also the list below).

**Last sentence of abstract may be cut into two.**

done.

**Relevant change list**

- Title change
- Added a paragraph at the end of **1 Introduction** to introduce the fact that the simulated L2 products are not exactly aligned with the actual specifications (lines 80-86).
- Added a new paragraph **2.3 L2 Product Technical Specifications**, regarding the technical specification on which the L2 product simulation is based, also to better explain the fact that the simulated L2 products are not exactly aligned with the actually foreseen specifications. Lines 140-159
- Added a table (Table 1) in **2.3 L2 Product Technical Specifications** to summarize the instrument parameters that are more relevant for the L2 simulation.
- Added three new references in paragraph 2.3: (ESA 2012a, ESA 2012b, EUMETSAT 2010), lines 146, 423-430.
- Added a paragraph at half **2.3 The CDF method** to explain the choice of $S_{coinc}$ (lines 176-180).

[revised manuscript text omitted]

---

## Author Response (AR3)

**Response to the associate editor**

**Unfortunately, the reviewer still does not recommend the publication of the manuscript in its present form. However, the reviewer does accept your turn towards a feasibility study (and I accept this as well). The main criticism of the reviewer is that in some places the manuscript still seems to refer to actual missions. I think that it should be feasible to meet the requirements of the reviewer without too much effort. I would like to encourage you to resubmit a corrected version, with the issues raised by the reviewer fixed. With these corrections in place, the paper will be acceptable.**

Dear associate editor, thanks for the message. Even if we only partially agree, we decided to meet all the revision requests also because this further increases the readability of the paper. Thanks for the essential contribution that you and both the reviewer gave improving the paper.

**Response to RC2**

**1) The name of the instruments can NOT be S4-TIR, S4-UV, S5-TIR and S5-UV. They must change to general names as GEO-TIR, GEO-UV, LEO-TIR and LEO-UV. Labeling in the whole manuscript and figures should be changed and S4, S5 should not be mentioned.**

Done both in the paper and in the supplementary material. Modified Figure 1, Figure 3, Figure 4, Figure 5, Figure 6, Figure 7, Figure S3, Figure S5, Figure S6, Figure S7, Table 1, Table 2, Table 3, Table 4.

**2) The sentence "according to the specifications of the Sentinel 4 and 5 missions of the Copernicus programme" in the abstract (line 30) should be withdrawn since it is not correct.**

Done.

**3) The sentence "according to the specifications of the atmospheric Sentinels" in line 74 should be changed to "according to specifications similar to those of the atmospheric Sentinels".**

Done.

**4) As explained before, the sentence (lines 89-90) "It is important to note that the specifications used to simulate L2 products were the state of the art at the time of their usage in 90 the AURORA project context, but are now out of date" is not correct. This should be removed.**

Done

**5) It should be clearly stated "The spatial sampling (spacing between pixels and shape) and in some cases the signal-to-noise ratio (the geostationary TIR instrument) of the instruments used in the present paper (both geostationary and low orbit ) are different from those of that will be onboard the Sentinel 4/MTG-S and EPS-SG".**

Done (lines 83-85)

**6) The sentence "These specifications are now partially outdated" in line 163 should be removed.**

Done

**7) The sentence "In table 1 and in the next sections of the paper, we will refer to UVNS/MetOp-SG as S5-UV1, to UVN/MTG as S4-UV1, to IASI-NG/MetOp-SG as S5-TIR and to IR/MTG as S4-TIR" should be modified and the name of the instruments CAN NOT use S4 and S5 since it creates confusion.**

Substituted with LEO-UV1, GEO-UV1, LEO-TIR and GEO-TIR.

**8) The sentence "S4-S5" in line 251 and also in Table 1 should be replaced by "GEO-LEO"… and that all through the manuscript.**

Done.

**9) Be aware that using IRS1b from Crevoisier et al 2014 for IRS/MTG-S is NOT correct. The SNR and spectral resolution of IRS1b are NOT those of IRS/MTG-S. This information creates confusion. Table 1 should only describe the GEO and LEO instruments in the present paper without introducing a confusion erroneously referring to actual satellite missions.**

We substituted S5 with LEO and S4 with GEO (see also point 7). We modified the Row tag "Instrument" in "Reference instrument" to refer to the actual satellite missions. We coherently modified the reference of Table 1 in the text (line 160).

[revised manuscript text omitted]
| S4:**GEO:TIR+UV1**_S5:**LEO:TIR+UV1** | Two or more GEOS4 pixels, one or more LEO-S5-TIR pixel, one or more LEO-S5-UV1 pixel. | 140 | 289.4 | 504 |
| S4:**GEO:TIR+UV1**_S5:**LEO:TIR** | Two or more GEOS4 pixels, one or more GEO-S5-TIR pixel, no LEO-S5-UV1 pixels. | 79 | 115.4 | 442 |
| S4:**GEO:TIR+UV1**_S5:**LEO:UV1** | Two or more GEOS4 pixels, one or more LEO-S5-UV1pixel, no LEO-S5-TIR pixels. | 0 | 0 | 0 |
| S5:**LEO:TIR+UV1** | No GEOS4 pixels, one or more LEO-S5-TIR pixels, one or more LEO-S5-UV1 pixels. | 142 | 26.2 | 71 |
| S5:**LEO:TIR** | No GEOS4 pixels, two or more LEO-S5-TIR pixels, no LEO-S5-UV1 pixels. | 60 | 8.9 | 26 |
| **TOTAL** | | 775 | 102.9 | 504 |

**Table 4: Like in Table2 but with a grid-box size of 1ºx1º. Ncells is the number of grid-boxes characterized by the considered FUS type; <NL2> is the mean number of individual L2 fusing profiles per grid-box and Max NL2is the maximum number of individual L2 fusing products per grid-box.**

[Figure]

**Figure 1: geographical distribution of the simulated L2 measurements and geo-location of the fused product. The black dash-dotted lines represent the borders of the 0.5°x0.625° grid cells.**

[Figure]

**Figure 2 (Left panel):** the absolute differences between L2 profiles and their true profiles (green lines), the absolute difference between the fused profile and the average of the true profiles (dark red continuous line), the average of σtotal of L2 simulations (black dash-dotted lines), σf total (dark red dash-dotted lines). **(Right panel):** the relative percentage differences between L2 profiles and their true profiles (green lines), the relative percentage difference between the fused profile and the average of the true profiles (dark red continuous line), the average of σtotal of L2 simulations normalized wrt the true profile and expressed in percentage (black dash-dotted lines), σf total normalized wrt the true profile and expressed in percentage (dark red dash-dotted lines).

[Figure]

[Figure]

**Figure.3 (Left panel): AKs diagonals.  GEO-TIR products (red lines),  LEO-TIR products (blue lines)  GEO-UV products (red lines) and FUS product (dark red line). In the text box, the average number of DOFs for each type of L2 product, the average number of DOFs for all L2 products and the number of DOFs of the FUS product are reported. (Right panel): Vertical resolution (FWHM) profiles.  GEO-TIR products (red lines),  LEO-TIR products (blue lines)  GEO-UV products (red lines) and FUS product (dark red line). In each panel, while the solid dark red line is a single one, the red and green lines are both 55 overlapped lines, and the blue lines are eight overlapped lines (one for each L2 product).**

500

[Figure]

[Figure]

**Figure 4 (Left panel): Rows of AK matrices at 6 km. (Right panel): Rows of AK matrices at 39 km  GEO-TIR products (red lines),  LEO-TIR products (blue lines)  GEO-UV products (red lines) and FUS product (dark red line).**

[Figure]

[Figure]

**Figure 5. Left panel: geographical distribution of FUS products differentiated by FUS type where the effect of the lower resolution of  LEO-UV1 respect to the other L2 products is the cause of the periodic FUS type transitions in the Mediterranean area. Right panel: histogram of the number of cells with a given number of L2 measurements differentiated by FUS type.**

[Figure]

[Figure]

**Figure 6: scatter plot of SF DOF as a function of the number of L2 measurements fused in each coincidence grid cell; different colours represent different FUS types.**

[Figure]

[Figure]

**Figure 7 (Left panel):** SF AK versus vertical level. **(Right panel):** SF ERR versus vertical level. In both panels, different colours of the symbols represent the FUS type, different sizes of the symbols represent the number of measurements that have been fused. The maximum symbol size shown in the legend corresponds to N=160.

505

---

## Author Response (AR4)

**Response to the associate editor**

**Dear authors, thank you for considering the changes suggested by the reviewer. The paper is almost ready for publication. However, I am afraid that the reviewer still would consider it as misleading to mention the Sentinel instruments. You have tried to solve the problem by changing the title from "Instrument" to "Reference instrument" but I do not think that the reader will know what the attribute "reference" is meant to tell them. In order to save time, I do not want another round of reviewing. Instead, I suggest that you simply add a table footnote to the row captions "Platform" and "Reference instrument", saying "The table entries in these rows are not meant to suggest that the specifications listed in this table refer directly to these instruments. Instead, the listed instruments are examples of instruments actually planned, with similar, however, slightly different specifications." With this final correction in place I will accept the paper.**

Dear associate editor, thanks for the message and for the contribute. We added the table footnote and corrected some typos and abbreviations (S5->LEO, S4->GEO, geostationary ->GEO, low orbit ->LEO).

**Relevant change list**

- See above

[revised manuscript text omitted]